# Finite-time Analysis of Approximate Policy Iteration for the Linear Quadratic Regulator

**Karl Krauth**
University of California, Berkeley
karlk@berkeley.edu

**Stephen Tu**
University of California, Berkeley
stephentu@berkeley.edu

**Benjamin Recht**
University of California, Berkeley
brecht@berkeley.edu

## Abstract

We study the sample complexity of approximate policy iteration (PI) for the Linear Quadratic Regulator (LQR), building on a recent line of work using LQR as a testbed to understand the limits of reinforcement learning (RL) algorithms on continuous control tasks. Our analysis quantifies the tension between policy improvement and policy evaluation, and suggests that policy evaluation is the dominant factor in terms of sample complexity. Specifically, we show that to obtain a controller that is within $\varepsilon$ of the optimal LQR controller, each step of policy evaluation requires at most $(n + d)^3/\varepsilon^2$ samples, where $n$ is the dimension of the state vector and $d$ is the dimension of the input vector. On the other hand, only $\log(1/\varepsilon)$ policy improvement steps suffice, resulting in an overall sample complexity of $(n + d)^3\varepsilon^{-2}\log(1/\varepsilon)$. We furthermore build on our analysis and construct a simple adaptive procedure based on $\varepsilon$-greedy exploration which relies on approximate PI as a sub-routine and obtains $T^{2/3}$ regret, improving upon a recent result of Abbasi-Yadkori et al. [3].

## 1 Introduction

With the recent successes of reinforcement learning (RL) on continuous control tasks, there has been a renewed interest in understanding the sample complexity of RL methods. A recent line of work has focused on the Linear Quadratic Regulator (LQR) as a testbed to understand the behavior and trade-offs of various RL algorithms in the continuous state and action space setting. These results can be broadly grouped into two categories: (1) the study of *model-based* methods which use data to build an estimate of the transition dynamics, and (2) *model-free* methods which directly estimate the optimal feedback controller from data without building a dynamics model as an intermediate step. Much of the recent progress in LQR has focused on the model-based side, with an analysis of robust control from Dean et al. [12] and certainty equivalence control by Fiechter [17] and Mania et al. [26]. These techniques have also been extended to the online, adaptive setting [1, 4, 11, 13, 31]. On the other hand, for classic model-free RL algorithms such as Q-learning, SARSA, and approximate policy iteration (PI), our understanding is much less complete within the context ofd LQR. This is despite the fact that these algorithms are well understood in the tabular (finite state and action space) setting. Indeed, most of the model-free analysis for LQR [16, 24, 35] has focused exclusively on derivative-free random search methods.

In this paper, we extend our understanding of model-free algorithms for LQR by studying the performance of approximate PI on LQR, which is a classic approximate dynamic programming algorithm. Approximate PI is a model-free algorithm which iteratively uses trajectory data to estimate

the state-value function associated to the current policy (via e.g. temporal difference learning), and then uses this estimate to greedily improve the policy. A key issue in analyzing approximate PI is to understand the trade-off between the number of policy improvement iterations, and the amount of data to collect for each policy evaluation phase. Our analysis quantifies this trade-off, showing that if least-squares temporal difference learning (LSTD-Q) [9, 20] is used for policy evaluation, then a trajectory of length $\widetilde{O}((n + d)^3/\varepsilon^2)$ for each inner step of policy evaluation combined with $\mathcal{O}(\log(1/\varepsilon))$ outer steps of policy improvement suffices to learn a controller that has $\varepsilon$-error from the optimal controller. This yields an overall sample complexity of $\mathcal{O}((n + d)^3 \varepsilon^{-2} \log(1/\varepsilon))$. Prior to our work, the only known guarantee for approximate PI on LQR was the asymptotic consistency result of Bradtke [10] in the setting of no process noise.

We also extend our analysis of approximate PI to the online, adaptive LQR setting popularized by Abbasi-Yadkori and Szepesvári [1]. By using a greedy exploration scheme similar to Dean et al. [13] and Mania et al. [26], we prove a $\widetilde{O}(T^{2/3})$ regret bound for a simple adaptive policy improvement algorithm. While the $T^{2/3}$ rate is sub-optimal compared to the $T^{1/2}$ regret from model-based methods [1, 11, 26], our analysis improves the $\widetilde{O}(T^{2/3+\varepsilon})$ regret (for $T \geq C^{1/\varepsilon}$) from the model-free Follow the Leader (FTL) algorithm of Abbasi-Yadkori et al. [3]. To the best of our knowledge, we give the best regret guarantee known for a model-free algorithm. We leave open the question of whether or not a model-free algorithm can achieve optimal $T^{1/2}$ regret.

## 2 Main Results

In this paper, we consider the following linear dynamical system:

$$x_{t+1} = Ax_t + Bu_t + w_t \,, \ \ w_t \sim \mathcal{N}(0, \sigma_w^2 I) \,, \ \ x_0 \sim \mathcal{N}(0, \Sigma_0) \,. \tag{2.1}$$

We let $n$ denote the dimension of the state $x_t$ and $d$ denote the dimension of the input $u_t$. For simplicity we assume that $d \leq n$, e.g. the system is under-actuated. We fix two positive definite cost matrices $(S, R)$, and consider the infinite horizon average-cost Linear Quadratic Regulator (LQR):

$$J_\star := \min_{\{u_t(\cdot)\}} \lim_{T \to \infty} \mathbb{E} \left[ \frac{1}{T} \sum_{t=1}^{T} x_t^\mathsf{T} S x_t + u_t^\mathsf{T} R u_t \right] \ \text{ subject to } (2.1) \,. \tag{2.2}$$

We assume the dynamics matrices $(A, B)$ are unknown to us, and our method of interaction with (2.1) is to choose an input sequence $\{u_t\}$ and observe the resulting states $\{x_t\}$.

We study the solution to (2.2) using *least-squares policy iteration (LSPI)*, a well-known approximate dynamic programming method in RL introduced by Lagoudakis and Parr [20]. The study of approximate PI on LQR dates back to the Ph.D. thesis of Bradtke [10], where he showed that for *noiseless* LQR (when $w_t = 0$ for all $t$), the approximate PI algorithm is asymptotically consistent. In this paper we expand on this result and quantify non-asymptotic rates for approximate PI on LQR. Proofs of all results can be found in the extended version of this paper [19].

**Notation.** For a positive scalar $x > 0$, we let $x_+ = \max\{1, x\}$. A square matrix $L$ is called stable if $\rho(L) < 1$ where $\rho(\cdot)$ denotes the spectral radius of $L$. For a symmetric matrix $M \in \mathbb{R}^{n \times n}$, we let $\mathsf{dlyap}(L, M)$ denote the unique solution to the discrete Lyapunov equation $P = L^\mathsf{T} P L + M$. We also let $\mathrm{svec}(M) \in \mathbb{R}^{n(n+1)/2}$ denote the vectorized version of the upper triangular part of $M$ so that $\|M\|_F^2 = \langle \mathrm{svec}(M), \mathrm{svec}(M) \rangle$. Finally, $\mathrm{smat}(\cdot)$ denotes the inverse of $\mathrm{svec}(\cdot)$, so that $\mathrm{smat}(\mathrm{svec}(M)) = M$.

### 2.1 Least-Squares Temporal Difference Learning (LSTD-Q)

The first component towards an understanding of approximate PI is to understand least-squares temporal difference learning (LSTD-Q) for $Q$-functions, which is the fundamental building block of LSPI. Given a deterministic policy $K_{\mathrm{eval}}$ which stabilizes $(A, B)$, the goal of LSTD-Q is to estimate the parameters of the $Q$-function associated to $K_{\mathrm{eval}}$. Bellman's equation for infinite-horizon average cost MDPs (c.f. Bertsekas [6]) states that the (relative) $Q$-function associated to a policy $\pi$ satisfies the following fixed-point equation:

$$\lambda + Q(x, u) = c(x, u) + \mathbb{E}_{x' \sim p(\cdot|x,u)}[Q(x', \pi(x'))] \,. \tag{2.3}$$

Here, $\lambda \in \mathbb{R}$ is a free parameter chosen so that the fixed-point equation holds. LSTD-Q operates under the *linear architecture* assumption, which states that the $Q$-function can be described as $Q(x, u) = q^{\mathsf{T}}\phi(x, u)$, for a known (possibly non-linear) feature map $\phi(x, u)$. It is well known that LQR satisfies the linear architecture assumption, since we have:

$$Q(x, u) = \text{svec}(Q)^{\mathsf{T}}\text{svec}\left(\begin{bmatrix} x \\ u \end{bmatrix}\begin{bmatrix} x \\ u \end{bmatrix}^{\mathsf{T}}\right), \quad Q = \begin{bmatrix} S & 0 \\ 0 & R \end{bmatrix} + \begin{bmatrix} A^{\mathsf{T}} \\ B^{\mathsf{T}} \end{bmatrix} V \begin{bmatrix} A & B \end{bmatrix},$$

$$V = \text{dlyap}(A + BK_{\text{eval}}, S + K_{\text{eval}}^{\mathsf{T}}RK_{\text{eval}}), \quad \lambda = \left\langle Q, \sigma_w^2 \begin{bmatrix} I \\ K_{\text{eval}} \end{bmatrix}\begin{bmatrix} I \\ K_{\text{eval}} \end{bmatrix}^{\mathsf{T}}\right\rangle.$$

Here, we slightly abuse notation and let $Q$ denote the $Q$-function and also the matrix parameterizing the $Q$-function. Now suppose that a trajectory $\{(x_t, u_t, x_{t+1})\}_{t=1}^{T}$ is collected. Note that LSTD-Q is an *off-policy* method (unlike the closely related LSTD estimator for value functions), and therefore the inputs $u_t$ can come from any sequence that provides sufficient excitation for learning. In particular, it does *not* have to come from the policy $K_{\text{eval}}$. In this paper, we will consider inputs of the form:

$$u_t = K_{\text{play}}x_t + \eta_t, \quad \eta_t \sim \mathcal{N}(0, \sigma_\eta^2 I), \tag{2.4}$$

where $K_{\text{play}}$ is a stabilizing controller for $(A, B)$. Once again we emphasize that $K_{\text{play}} \neq K_{\text{eval}}$ in general. Furthermore, the policy under $K_{\text{eval}}$ is stochastic while the policy under $K_{\text{play}}$ is stochastic, where the injected noise $\eta_t$ is needed in order to provide sufficient excitation for learning. In order to describe the LSTD-Q estimator, we define the following quantities which play a key role throughout the paper:

$$\phi_t := \phi(x_t, u_t), \quad \psi_t := \phi(x_t, K_{\text{eval}}x_t),$$

$$f := \text{svec}\left(\sigma_w^2 \begin{bmatrix} I \\ K_{\text{eval}} \end{bmatrix}\begin{bmatrix} I \\ K_{\text{eval}} \end{bmatrix}^{\mathsf{T}}\right), \quad c_t := x_t^{\mathsf{T}}Sx_t + u_t^{\mathsf{T}}Ru_t.$$

The LSTD-Q estimator estimates $q$ via:

$$\widehat{q} := \left(\sum_{t=1}^{T}\phi_t(\phi_t - \psi_{t+1} + f)^{\mathsf{T}}\right)^{\dagger}\sum_{t=1}^{T}\phi_t c_t. \tag{2.5}$$

Here, $(\cdot)^{\dagger}$ denotes the Moore-Penrose pseudo-inverse. Our first result establishes a non-asymptotic bound on the quality of the estimator $\widehat{q}$, measured in terms of $\|\widehat{q} - q\|$. Before we state our result, we introduce a key definition that we will use extensively.

**Definition 1.** *Let $L$ be a square matrix. Let $\tau \geq 1$ and $\rho \in (0, 1)$. We say that $L$ is $(\tau, \rho)$-stable if*

$$\|L^k\| \leq \tau\rho^k, \quad k = 0, 1, 2, \dots.$$

While stability of a matrix is an asymptotic notion, Definition 1 quantifies the degree of stability by characterizing the transient response of the powers of a matrix by the parameter $\tau$. It is closely related to the notion of *strong stability* from Cohen et al. [11].

With Definition 1 in place, we state our first result for LSTD-Q.

**Theorem 2.1.** *Fix a $\delta \in (0, 1)$. Let policies $K_{\text{play}}$ and $K_{\text{eval}}$ stabilize $(A, B)$, and assume that both $A + BK_{\text{play}}$ and $A + BK_{\text{eval}}$ are $(\tau, \rho)$-stable. Let the initial state $x_0 \sim \mathcal{N}(0, \Sigma_0)$ and consider the inputs $u_t = K_{\text{play}}x_t + \eta_t$ with $\eta_t \sim \mathcal{N}(0, \sigma_\eta^2 I)$. For simplicity, assume that $\sigma_\eta \leq \sigma_w$. Let $P_\infty$ denote the steady-state covariance of the trajectory $\{x_t\}$:*

$$P_\infty = \text{dlyap}((A + BK_{\text{play}})^{\mathsf{T}}, \sigma_w^2 I + \sigma_\eta^2 BB^{\mathsf{T}}). \tag{2.6}$$

*Define the proxy variance $\overline{\sigma}^2$ by:*

$$\overline{\sigma}^2 := \tau^2\rho^4\|\Sigma_0\| + \|P_\infty\| + \sigma_\eta^2\|B\|^2. \tag{2.7}$$

*Suppose that $T$ satisfies:*

$$T \geq \widetilde{O}(1)\max\left\{(n + d)^2, \frac{\tau^4}{\rho^4(1 - \rho^2)^2}\frac{(n + d)^4}{\sigma_\eta^4}\sigma_w^2\overline{\sigma}^2\|K_{\text{play}}\|_+^4\|K_{\text{eval}}\|_+^8(\|A\|^4 + \|B\|^4)_+\right\}. \tag{2.8}$$

*Then we have with probability at least $1 - \delta$,*

$$\|\widehat{q} - q\| \leq \widetilde{O}(1) \frac{\tau^2}{\rho^2(1-\rho^2)} \frac{(n+d)}{\sigma_\eta^2 \sqrt{T}} \sigma_w \overline{\sigma} \|K_{\text{play}}\|_+^2 \|K_{\text{eval}}\|_+^4 (\|A\|^2 + \|B\|^2)_+ \|Q^{K_{\text{eval}}}\|_F . \quad (2.9)$$

*Here the $\widetilde{O}(1)$ hides* $\text{polylog}(n, \tau, \|\Sigma_0\|, \|P_\infty\|, \|K_{\text{play}}\|, T/\delta, 1/\sigma_\eta)$ *factors.*

Theorem 2.1 states that:

$$T \leq \widetilde{O}\left((n+d)^4, \frac{1}{\sigma_\eta^4} \frac{(n+d)^3}{\varepsilon^2}\right)$$

timesteps are sufficient to achieve error $\|\widehat{q} - q\| \leq \varepsilon$ w.h.p. Several remarks are in order. First, while the $(n+d)^4$ burn-in is likely sub-optimal, the $(n+d)^3/\varepsilon^2$ dependence is sharp as shown by the asymptotic results of Tu and Recht [35]. Second, the $1/\sigma_\eta^4$ dependence on the injected excitation noise will be important when we study the online, adaptive setting in Section 2.3. We leave improving the polynomial dependence of the burn-in period to future work.

The proof of Theorem 2.1 rests on top of several recent advances. First, we build off the work of Abbasi-Yadkori et al. [3] to derive a new basic inequality for LSTD-Q which serves as a starting point for the analysis. Next, we combine the small-ball techniques of Simchowitz et al. [33] with the self-normalized martingale inequalities of Abbasi-Yadkori et al. [2]. While an analysis of LSTD-Q is presented in Abbasi-Yadkori et al. [3] (which builds on the analysis for LSTD from Tu and Recht [34]), a direct application of their result yields a $1/\sigma_\eta^8$ dependence; the use of self-normalized inequalities is necessary in order to reduce this dependence to $1/\sigma_\eta^4$.

## 2.2 Least-Squares Policy Iteration (LSPI)

With Theorem 2.1 in place, we are ready to present the main results for LSPI. We describe two versions of LSPI in Algorithm 1 and Algorithm 2.

---

**Algorithm 1** LSPIv1 for LQR

**Input:** $K_0$: initial stabilizing controller,
$\quad\quad\quad N$: number of policy iterations,
$\quad\quad\quad T$: length of rollout,
$\quad\quad\quad \sigma_\eta^2$: exploration variance,
$\quad\quad\quad \mu$: lower eigenvalue bound.
1: Collect $\mathcal{D} = \{(x_k, u_k, x_{k+1})\}_{k=1}^T$ with input $u_k = K_0 x_k + \eta_k, \eta_k \sim \mathcal{N}(0, \sigma_\eta^2 I)$.
2: **for** $t = 0, ..., N-1$ **do**
3: $\quad \widehat{Q}_t = \text{Proj}_\mu(\text{LSTDQ}(\mathcal{D}, K_t))$.
4: $\quad K_{t+1} = G(\widehat{Q}_t)$. [See (2.10).]
5: **end for**
6: **return** $K_N$.

---

**Algorithm 2** LSPIv2 for LQR

**Input:** $K_0$: initial stabilizing controller,
$\quad\quad\quad N$: number of policy iterations,
$\quad\quad\quad T$: length of rollout,
$\quad\quad\quad \sigma_\eta^2$: exploration variance,
$\quad\quad\quad \mu$: lower eigenvalue bound.
1: **for** $t = 0, ..., N-1$ **do**
2: $\quad$ Collect $\mathcal{D}_t = \{(x_k^{(t)}, u_k^{(t)}, x_{k+1}^{(t)})\}_{k=1}^T$,
$\quad\quad u_k^{(t)} = K_0 x_k^{(t)} + \eta_k^{(t)}, \eta_k^{(t)} \sim \mathcal{N}(0, \sigma_\eta^2 I)$.
3: $\quad \widehat{Q}_t = \text{Proj}_\mu(\text{LSTDQ}(\mathcal{D}, K_t))$.
4: $\quad K_{t+1} = G(\widehat{Q}_t)$.
5: **end for**
6: **return** $K_N$.

---

In Algorithms 1 and 2, $\text{Proj}_\mu(\cdot) = \arg\min_{X=X^\mathsf{T}:X\succeq \mu \cdot I}\|X - \cdot\|_F$ is the Euclidean projection onto the set of symmetric matrices lower bounded by $\mu \cdot I$. Furthermore, the map $G(\cdot)$ takes an $(n+d) \times (n+d)$ positive definite matrix and returns a $d \times n$ matrix:

$$G\left(\begin{bmatrix} Q_{11} & Q_{12} \\ Q_{12}^\mathsf{T} & Q_{22} \end{bmatrix}\right) = -Q_{22}^{-1} Q_{12}^\mathsf{T} . \quad (2.10)$$

Algorithm 1 corresponds to the version presented in Lagoudakis and Parr [20], where all the data $\mathcal{D}$ is collected up front and is re-used in every iteration of LSTD-Q. Algorithm 2 is the one we will analyze in this paper, where new data is collected for every iteration of LSTD-Q. The modification made in Algorithm 2 simplifies the analysis by allowing the controller $K_t$ to be independent of the data $\mathcal{D}_t$ in LSTD-Q. We remark that this does *not* require the system to be reset after every iteration of LSTD-Q. We leave analyzing Algorithm 1 to future work.

Before we state our main finite-sample guarantee for Algorithm 2, we review the notion of a (relative) value-function. Similarly to (relative) $Q$-functions, the infinite horizon average-cost Bellman equation states that the (relative) value function $V$ associated to a policy $\pi$ satisfies the fixed-point equation:

$$\lambda + V(x) = c(x, \pi(x)) + \mathbb{E}_{x' \sim p(\cdot|x,\pi(x))}[V(x')]. \tag{2.11}$$

For a stabilizing policy $K$, it is well known that for LQR the value function $V(x) = x^{\mathsf{T}} V x$ with

$$V = \mathsf{dlyap}(A + BK, S + K^{\mathsf{T}} RK), \;\; \lambda = \langle \sigma_w^2 I, V \rangle.$$

Once again as we did for $Q$-functions, we slightly abuse notation and let $V$ denote the value function and the matrix that parameterizes the value function. Our main result for Algorithm 2 appears in the following theorem. For simplicity, we will assume that $\|S\| \geq 1$ and $\|R\| \geq 1$.

**Theorem 2.2.** *Fix a $\delta \in (0,1)$. Let the initial policy $K_0$ input to Algorithm 2 stabilize $(A, B)$. Suppose the initial state $x_0 \sim \mathcal{N}(0, \Sigma_0)$ and that the excitation noise satisfies $\sigma_\eta \leq \sigma_w$. Recall that the steady-state covariance of the trajectory $\{x_t\}$ is*

$$P_\infty = \mathsf{dlyap}((A + BK_0)^{\mathsf{T}}, \sigma_w^2 I + \sigma_\eta^2 BB^{\mathsf{T}}).$$

*Let $V_0$ denote the value function associated to the initial policy $K_0$, and $V_\star$ denote the value function associated to the optimal policy $K_\star$ for the LQR problem (2.2). Define the variables $\mu, L$ as:*

$$\mu := \min\{\lambda_{\min}(S), \lambda_{\min}(R)\},$$
$$L := \max\{\|S\|, \|R\|\} + 2(\|A\|^2 + \|B\|^2 + 1)\|V_0\|_+.$$

*Fix an $\varepsilon > 0$ that satisfies:*

$$\varepsilon \leq 5\left(\frac{L}{\mu}\right)^2 \min\left\{1, \frac{2\log(\|V_0\|/\lambda_{\min}(V_\star))}{e}, \frac{\|V_\star\|^2}{8\mu^2 \log(\|V_0\|/\lambda_{\min}(V_\star))}\right\}. \tag{2.12}$$

*Suppose we run Algorithm 2 for $N := N_0 + 1$ policy improvement iterations where*

$$N_0 := \left\lceil (1 + L/\mu) \log\left(\frac{2\log(\|V_0\|/\lambda_{\min}(V_\star))}{\varepsilon}\right)\right\rceil, \tag{2.13}$$

*and we set the rollout length $T$ to satisfy:*

$$T \geq \widetilde{O}(1) \max\left\{ (n+d)^2, \right.$$

$$\frac{L^2}{(1 - \mu/L)^2}\left(\frac{L}{\mu}\right)^{17} \frac{(n+d)^4}{\sigma_\eta^4} \sigma_w^2(\|\Sigma_0\| + \|P_\infty\| + \sigma_\eta^2\|B\|^2),$$

$$\left. \frac{1}{\varepsilon^2} \frac{L^4}{(1 - \mu/L)^2}\left(\frac{L}{\mu}\right)^{42} \frac{(n+d)^3}{\sigma_\eta^4} \sigma_w^2(\|\Sigma_0\| + \|P_\infty\| + \sigma_\eta^2\|B\|^2)\right\}. \tag{2.14}$$

*Then with probability $1 - \delta$, we have that each policy $K_t$ for $t = 1, ..., N$ stabilizes $(A, B)$ and furthermore:*

$$\|K_N - K_\star\| \leq \varepsilon.$$

*Here the $\widetilde{O}(1)$ hides* $\mathrm{polylog}(n, \tau, \|\Sigma_0\|, \|P_\infty\|, L/\mu, T/\delta, N_0, 1/\sigma_\eta)$ *factors.*

Theorem 2.2 states roughly that $T \cdot N \leq \widetilde{O}(\frac{(n+d)^3}{\varepsilon^2} \log(1/\varepsilon))$ samples are sufficient for LSPI to recover a controller $K$ that is within $\varepsilon$ of the optimal $K_\star$. That is, only $\log(1/\varepsilon)$ iterations of policy improvement are necessary, and furthermore more iterations of policy improvement do not necessary help due to the inherent statistical noise of estimating the $Q$-function for every policy $K_t$. We note that the polynomial factor in $L/\mu$ is by no means optimal and was deliberately made quite conservative in order to simplify the presentation of the bound. A sharper bound can be recovered from our analysis techniques at the expense of a less concise expression.

It is worth taking a moment to compare Theorem 2.2 to classical results in the RL literature regarding approximate policy iteration. For example, a well known result (c.f. Theorem 7.1 of Lagoudakis and

Parr [20]) states that if LSTD-Q is able to return $Q$-function estimates with error $L_\infty$ bounded by $\varepsilon$ at every iteration, then letting $\widehat{Q}_t$ denote the approximate $Q$-function at the $t$-th iteration of LSPI:

$$\limsup_{t\to\infty} \|\widehat{Q}_t - Q_\star\|_\infty \le \frac{2\gamma\varepsilon}{(1-\gamma)^2} \,. \tag{2.15}$$

Here, $\gamma$ is the discount factor of the MDP. Theorem 2.2 is qualitatively similar to this result in that we show roughly that $\varepsilon$ error in the $Q$-function estimate translates to $\varepsilon$ error in the estimated policy. However, there are several fundamental differences. First, our analysis does not rely on discounting to show contraction of the Bellman operator. Instead, we use the $(\tau, \rho)$-stability of closed loop system to achieve this effect. Second, our analysis does not rely on $L_\infty$ bounds on the estimated $Q$-function, although we remark that similar type of results to (2.15) exist also in $L_p$ (see e.g. [27, 29]). Finally, our analysis is non-asymptotic.

The proof of Theorem 2.2 combines the estimation guarantee of Theorem 2.1 with a new analysis of policy iteration for LQR, which we believe is of independent interest. Our new policy iteration analysis combines the work of Bertsekas [7] on policy iteration in infinite horizon average cost MDPs with the contraction theory of Lee and Lim [22] for non-linear matrix equations.

## 2.3 LSPI for Adaptive LQR

We now turn our attention to the online, adaptive LQR problem as studied in Abbasi-Yadkori and Szepesvári [1]. In the adaptive LQR problem, the quantity of interest is the *regret*, defined as:

$$\mathsf{Regret}(T) := \sum_{t=1}^{T} x_t^\mathsf{T} S x_t + u_t^\mathsf{T} R u_t - T \cdot J_\star \,. \tag{2.16}$$

Here, the algorithm is penalized for the cost incurred from learning the optimal policy $K_\star$, and must balance exploration (to better learn the optimal policy) versus exploitation (to reduce cost). As mentioned previously, there are several known algorithms which achieve $\widetilde{O}(\sqrt{T})$ regret [1, 4, 11, 26, 31]. However, these algorithms operate in a *model-based* manner, using the collected data to build a confidence interval around the true dynamics $(A, B)$. On the other hand, the performance of adaptive algorithms which are *model-free* is less well understood. We use the results of the previous section to give an adaptive model-free algorithm for LQR which achieves $\widetilde{O}(T^{2/3})$ regret, which improves upon the $\widetilde{O}(T^{2/3+\varepsilon})$ regret (for $T \ge C^{1/\varepsilon}$) achieved by the adaptive model-free algorithm of Abbasi-Yadkori et al. [3]. Our adaptive algorithm based on LSPI is shown in Algorithm 3.

---

**Algorithm 3** Online Adaptive Model-free LQR Algorithm

---

**Input:** Initial stabilizing controller $K^{(0)}$, number of epochs $E$, epoch multiplier $T_{\mathrm{mult}}$, lower eigenvalue bound $\mu$.
1: **for** $i = 0, ..., E-1$ **do**
2:     Set $T_i = T_{\mathrm{mult}} 2^i$.
3:     Set $\sigma_{\eta,i}^2 = \sigma_w^2 \left(\frac{1}{2^i}\right)^{1/3}$.
4:     Set $K^{(i+1)} = \mathsf{LSPIv2}(K_0 = K^{(i)}, N = \widetilde{O}((i+1)\Gamma_\star/\mu), T = T_i, \sigma_\eta^2 = \sigma_{\eta,i}^2)$.
5: **end for**

---

Using an analysis technique similar to that in Dean et al. [13], we prove the following $\widetilde{O}(T^{2/3})$ regret bound for Algorithm 3.

**Theorem 2.3.** *Fix a $\delta \in (0,1)$. Let the initial feedback $K^{(0)}$ stabilize $(A, B)$ and let $V^{(0)}$ denote its associated value function. Also let $K_\star$ denote the optimal LQR controller and let $V_\star$ denote the optimal value function. Let $\Gamma_\star = 1 + \max\{\|A\|, \|B\|, \|V^{(0)}\|, \|V_\star\|, \|K^{(0)}\|, \|K_\star\|, \|Q\|, \|R\|\}$. Suppose that $T_{\mathrm{mult}}$ is set to:*

$$T_{\mathrm{mult}} \ge \widetilde{O}(1)\mathrm{poly}(\Gamma_\star, n, d, 1/\lambda_{\min}(S)) \,.$$

*and suppose $\mu$ is set to $\mu = \min\{\lambda_{\min}(S), \lambda_{\min}(R)\}$. With probability at least $1 - \delta$, we have that the regret of Algorithm 3 satisfies:*

$$\mathsf{Regret}(T) \le \widetilde{O}(1)\mathrm{poly}(\Gamma_\star, n, d, 1/\lambda_{\min}(S))T^{2/3} \,.$$

We note that the regret scaling as $T^{2/3}$ in Theorem 2.3 is due to the $1/\sigma_\eta^4$ dependence from LSTD-Q (c.f. (2.9)). As mentioned previously, the existing LSTD-Q analysis from Abbasi-Yadkori et al. [3] yields a $1/\sigma_\eta^8$ dependence in LSTD-Q; using this $1/\sigma_\eta^8$ dependence in the analysis of Algorithm 3 would translate into $T^{4/5}$ regret.

## 3 Related Work

For model-based methods, in the offline setting Fiechter [17] provided the first PAC-learning bound for infinite horizon *discounted* LQR using certainty equivalence (nominal) control. Later, Dean et al. [12] use tools from robust control to analyze a robust synthesis method for infinite horizon *average cost* LQR, which is applicable in regimes of moderate uncertainty when nominal control fails. Mania et al. [26] show that certainty equivalence control actually provides a fast $\mathcal{O}(\varepsilon^2)$ rate of sub-optimality where $\varepsilon$ is the size of the parameter error, unlike the $\mathcal{O}(\varepsilon)$ sub-optimality guarantee of [12, 17]. For the online adaptive setting, [1, 4, 11, 18, 26] give $\widetilde{O}(\sqrt{T})$ regret algorithms. A key component of model-based algorithms is being able to quantify a confidence interval for the parameter estimate, for which several recent works [14, 32, 33] provide non-asymptotic results.

Turning to model-free methods, Tu and Recht [34] study the behavior of least-squares temporal difference (LSTD) for learning the *discounted value* function associated to a stabilizing policy. They evaluate the LSPI algorithm studied in this paper empirically, but do not provide any analysis. In terms of policy optimization, most of the work has focused on derivative-free random search methods [16, 24]. Tu and Recht [35] study a special family of LQR instances and characterize the asymptotic behavior of both model-based certainty equivalent control versus policy gradients (REINFORCE), showing that policy gradients has polynomially worse sample complexity. Most related to our work is Abbasi-Yadkori et al. [3], who analyze a model-free algorithm for adaptive LQR based on ideas from online convex optimization. LSTD-Q is a sub-routine of their algorithm, and their analysis incurs a sub-optimal $1/\sigma_\eta^8$ dependence on the injected exploration noise, which we improve to $1/\sigma_\eta^4$ using self-normalized martingale inequalities [2]. This improvement allows us to use a simple greedy exploration strategy to obtain $T^{2/3}$ regret. Finally, as mentioned earlier, the Ph.D. thesis of Bradtke [10] presents an asymptotic consistency argument for approximate PI for discounted LQR in the noiseless setting (i.e. $w_t = 0$ for all $t$).

For the general function approximation setting in RL, Antos et al. [5] and Lazaric et al. [21] analyze variants of LSPI for discounted MDPs where the state space is compact and the action space finite. In Lazaric et al. [21], the policy is greedily updated via an update operator that requires access to the underlying dynamics (and is therefore not implementable). Farahmand et al. [15] extend the results of Lazaric et al. [21] to when the function spaces considered are reproducing kernel Hilbert spaces. Zou et al. [37] give a finite-time analysis of both Q-learning and SARSA, combining the asymptotic analysis of Melo et al. [28] with the finite-time analysis of TD-learning from Bhandari et al. [8]. We note that checking the required assumptions to apply the results of Zou et al. [37] is non-trivial (c.f. Section 3.1, [28]). We are un-aware of any non-asymptotic analysis of LSPI in the *average cost* setting, which is more difficult as the Bellman operator is no longer a contraction.

Finally, we remark that our LSPI analysis relies on understanding exact policy iteration for LQR, which is closely related to the fixed-point Riccati recurrence (value iteration). An elegant analysis for value iteration is given by Lincoln and Rantzer [23]. Recently, Fazel et al. [16] show that exact policy iteration is a special case of Gauss-Newton and prove linear convergence results. Our analysis, on the other hand, is based on combining the fixed-point theory from Lee and Lim [22] with recent work on policy iteration for average cost problems from Bertsekas [7].

## 4 Experiments

We first look at the performance of LSPI in the non-adaptive setting (Section 2.2). Here, we compare LSPI to other popular model-free methods, and the model-based certainty equivalence (nominal) controller (c.f. [26]). For model-free, we look at policy gradients (REINFORCE) (c.f. [36]) and derivative-free optimization (c.f. [24, 25, 30]). A full description of the methods we compare to is given in the full paper [19].

We consider the LQR instance $(A, B, S, R)$ with $A = \begin{bmatrix} 0.95 & 0.01 & 0 \\ 0.01 & 0.95 & 0.01 \\ 0 & 0.01 & 0.95 \end{bmatrix}$, $B = \begin{bmatrix} 1 & 0.1 \\ 0 & 0.1 \\ 0 & 0.1 \end{bmatrix}$, $S = I_3$,

and $R = I_2$. We choose an LQR problem where the $A$ matrix is stable, since the model-free methods we consider need to be seeded with an initial stabilizing controller; using a stable $A$ allows us to start at $K_0 = 0_{2 \times 3}$. We fix the process noise $\sigma_w = 1$. The model-based nominal method learns $(A, B)$ using least-squares, exciting the system with Gaussian inputs $u_t$ with variance $\sigma_u = 1$.

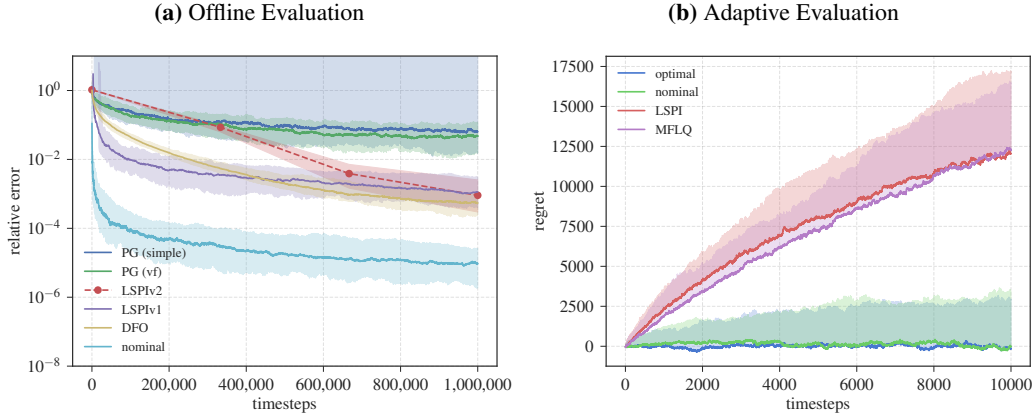

**Figure 1:** The performance of various model-free methods compared with the nominal controller. **(a)** Plot of non-adaptive performance. The shaded regions represent the lower 10th and upper 90th percentile over 100 trials, and the solid line represents the median performance. Here, PG (simple) is policy gradients with the simple baseline, PG (vf) is policy gradients with the value function baseline, LSPIv2 is Algorithm 2, LSPIv1 is Algorithm 1, and DFO is derivative-free optimization. **(b)** Plot of adaptive performance. The shaded regions represent the median to upper 90th percentile over 100 trials. Here, LSPI is Algorithm 3 using LSPIv1, MFLQ is from Abbasi-Yadkori et al. [3], nominal is the $\varepsilon$-greedy adaptive certainty equivalent controller (c.f. [13]), and optimal has access to the true dynamics.

For policy gradients and derivative-free optimization, we use the projected stochastic gradient descent (SGD) method with a constant step size $\mu$ as the optimization procedure. For policy iteration, we evaluate both LSPIv1 (Algorithm 1) and LSPIv2 (Algorithm 2). For every iteration of LSTD-Q, we project the resulting $Q$-function parameter matrix onto the set $\{Q : Q \succeq \gamma I\}$ with $\gamma = \min\{\lambda_{\min}(S), \lambda_{\min}(R)\}$. For LSPIv1, we choose $N = 15$ by picking the $N \in [5, 10, 15]$ which results in the best performance after $T = 10^6$ timesteps. For LSPIv2, we set $(N, T) = (3, 333333)$ which yields the lowest cost over the grid $N \in [1, 2, 3, 4, 5, 6, 7]$ and $T$ such that $NT = 10^6$.

Next, we compare the performance of LSPI in the adaptive setting (Section 2.3). We compare LSPI against the model-free linear quadratic control (MFLQ) algorithm of Abbasi-Yadkori et al. [3], nominal certainty equivalence controller (c.f. [13]), and the optimal controller. We use the example

of Dean et al. [12], with $A = \begin{bmatrix} 1.01 & 0.01 & 0 \\ 0.01 & 1.01 & 0.01 \\ 0 & 0.01 & 1.01 \end{bmatrix}$, $B = I$, $S = 10I_3$, $R = I_3$, and $\sigma_w = 1$.

Figure 1 shows the results of these experiments. In Figure 1a, we plot the relative error $(J(\widehat{K}) - J_\star)/J_\star$ versus the number of timesteps. We see that the model-based certainty equivalence (nominal) method is more sample efficient than the other model-free methods considered. We also see that the value function baseline is able to dramatically reduce the variance of the policy gradient estimator compared to the simple baseline. The DFO method performs the best out of all the model-free methods considered on this example after $10^6$ timesteps, although the performance of policy iteration is comparable. In Figure 1b, we plot the regret (c.f. Equation 2.16). We see that LSPI and MFLQ both perform similarly with MFLQ slightly outperforming LSPI. We also note that the model-based nominal methods performs significantly better than both LSPI and MFLQ.

## 5 Conclusion

We studied the sample complexity of approximate PI on LQR, showing that roughly $(n + d)^3 \varepsilon^{-2} \log(1/\varepsilon)$ samples are sufficient to estimate a controller that is within $\varepsilon$ of the optimal. We also show how to turn this offline method into an adaptive LQR method with $T^{2/3}$ regret. Several questions remain open with our work. The first is if policy iteration is able to achieve $T^{1/2}$ regret, which is possible with other model-based methods. The second is whether or not model-free methods provide advantages in situations of partial observability for LQ control. Finally, an asymptotic analysis of LSPI, in the spirit of Tu and Recht [35], is of interest in order to clarify which parts of our analysis are sub-optimal due to the techniques we use versus are inherent in the algorithm.

## Acknowledgments

We thank the anonymous reviewers for their valuable feedback. We also thank the authors of Abbasi-Yadkori et al. [3] for providing us with an implementation of their model-free LQ algorithm. ST is supported by a Google PhD fellowship. This work is generously supported in part by ONR awards N00014-17-1-2191, N00014-17-1-2401, and N00014-18-1-2833, the DARPA Assured Autonomy (FA8750-18-C-0101) and Lagrange (W911NF-16-1-0552) programs, a Siemens Futuremakers Fellowship, and an Amazon AWS AI Research Award.

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
