[Supplementary Material · paper-4597-full.pdf]

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

}$. Recall that we are interested in finding the $Q$ function for a given policy $K_{\mathrm{eval}}$, and we have defined the vectors:

$$\phi_t = \phi(x_t, u_t)\,, \ \ \psi_t = \phi(x_t, K_{\mathrm{eval}}x_t)\,,$$

$$f = \mathrm{svec}\left(\sigma_w^2 \begin{bmatrix} I \\ K_{\mathrm{eval}} \end{bmatrix} \begin{bmatrix} I \\ K_{\mathrm{eval}} \end{bmatrix}^{\mathsf{T}}\right)\,, \ \ c_t = x_t^{\mathsf{T}} S x_t + u_t^{\mathsf{T}} R u_t\,.$$

Also recall that the input sequence $u_t$ being played is given by $u_t = K_{\mathrm{play}} x_t + \eta_t$, with $\eta_t \sim \mathcal{N}(0, \sigma_\eta^2 I)$. Both policies $K_{\mathrm{eval}}$ and $K_{\mathrm{play}}$ are assumed to stabilize $(A, B)$. Because of stability, we have that $P_t$ converges to a limit $P_\infty = \mathsf{dlyap}((A + BK_{\mathrm{play}})^{\mathsf{T}}, \sigma_w^2 I + \sigma_\eta^2 BB^{\mathsf{T}})$, where $P_t$ is:

$$P_t := \sum_{k=0}^{t-1} (A + BK_{\mathrm{play}})^k (\sigma_w^2 I + \sigma_\eta BB^{\mathsf{T}})((A + BK_{\mathrm{play}})^{\mathsf{T}})^k\,.$$

The covariance of $x_t$ for $t \geq 1$ is:

$$\mathrm{Cov}(x_t) = \Sigma_t := P_t + (A + BK_{\mathrm{play}})^t \Sigma_0 ((A + BK_{\mathrm{play}})^{\mathsf{T}})^t\,.$$

We define the following data matrices:

$$\Phi = \begin{bmatrix} -\phi_1^{\mathsf{T}}- \\ \vdots \\ -\phi_T^{\mathsf{T}}- \end{bmatrix}\,, \ \ \Psi_+ = \begin{bmatrix} -\psi_2^{\mathsf{T}}- \\ \vdots \\ -\psi_{T+1}^{\mathsf{T}}- \end{bmatrix}\,, \ \ c = (c_1, ..., c_T)^{\mathsf{T}}\,, \ \ F = \begin{bmatrix} -f^{\mathsf{T}}- \\ \vdots \\ -f^{\mathsf{T}}- \end{bmatrix}\,.$$

With this notation, the LSTD-Q estimator is:

$$\widehat{q} = \left(\Phi^{\mathsf{T}}(\Phi - \Psi_+ + F)\right)^{\dagger} \Phi^{\mathsf{T}} c\,.$$

Next, let $\Xi$ be the matrix:

$$\Xi = \begin{bmatrix} -\mathbb{E}[\phi(x_2, K_{\mathrm{eval}}x_2)|x_1, u_1]^{\mathsf{T}}- \\ \vdots \\ -\mathbb{E}[\phi(x_{T+1}, K_{\mathrm{eval}}x_{T+1})|x_T, u_T]^{\mathsf{T}}- \end{bmatrix}\,.$$

For what follows, we let the notation $\otimes_s$ denote the *symmetric* Kronecker product. See **?** ] for more details. The following lemma gives us a starting point for analysis. It is based on Lemma 4.1 of Abbasi-Yadkori et al. [3]. Recall that $q = \mathrm{svec}(Q)$ and $Q$ is the matrix which parameterizes the $Q$-function for $K_{\mathrm{eval}}$.

**Lemma A.1** (Lemma 4.1, [3]). *Let $L := \begin{bmatrix} I \\ K_{\mathrm{eval}} \end{bmatrix} [A \ \ B]$. Suppose that $\Phi$ has full column rank, and that*

$$\frac{\|(\Phi^{\mathsf{T}}\Phi)^{-1/2}\Phi^{\mathsf{T}}(\Xi - \Psi_+)\|}{\sigma_{\min}(\Phi)\sigma_{\min}(I - L \otimes_s L)} \leq 1/2\,.$$

*Then we have:*

$$\|\widehat{q} - q\| \leq 2\frac{\|(\Phi^{\mathsf{T}}\Phi)^{-1/2}\Phi^{\mathsf{T}}(\Xi - \Psi_+)q\|}{\sigma_{\min}(\Phi)\sigma_{\min}(I - L \otimes_s L)}\,. \tag{A.1}$$

*Proof.* By the Bellman equation (2.3), we have the identity:

$$\Phi q = c + (\Xi - F)q$$

By the definition of $\widehat{q}$, we have the identity:

$$\Phi\widehat{q} = P_\Phi(c + (\Psi_+ - F)\widehat{q})\,,$$

where $P_\Phi = \Phi(\Phi^{\mathsf{T}}\Phi)^{-1}\Phi^{\mathsf{T}}$ is the orthogonal projector onto the columns of $\Phi$. Combining these two identities gives us:

$$P_\Phi(\Phi - \Xi + F)(q - \widehat{q}) = P_\Phi(\Xi - \Psi_+)\widehat{q}\,.$$

Next, the $i$-th row of $\Phi - \Xi + F$ is:

$$\text{svec}\left( \begin{bmatrix} x_i \\ u_i \end{bmatrix} \begin{bmatrix} x_i \\ u_i \end{bmatrix}^{\mathsf{T}} - \mathbb{E}\left[ \begin{bmatrix} I \\ K_{\text{eval}} \end{bmatrix} \tilde{x}\tilde{x}^{\mathsf{T}} \begin{bmatrix} I \\ K_{\text{eval}} \end{bmatrix}^{\mathsf{T}} \,\middle|\, x_i, u_i \right] + \sigma_w^2 \begin{bmatrix} I \\ K_{\text{eval}} \end{bmatrix} \begin{bmatrix} I \\ K_{\text{eval}} \end{bmatrix}^{\mathsf{T}} \right)$$

$$= \text{svec}\left( \begin{bmatrix} x_i \\ u_i \end{bmatrix} \begin{bmatrix} x_i \\ u_i \end{bmatrix}^{\mathsf{T}} - L \begin{bmatrix} x_i \\ u_i \end{bmatrix} \begin{bmatrix} x_i \\ u_i \end{bmatrix}^{\mathsf{T}} L^{\mathsf{T}} \right)$$

$$= (I - L \otimes_s L)\phi(x_i, u_i)\,,$$

where $\tilde{x} = Ax_i + Bu_i + w_i$. Therefore, $\Phi - \Xi + F = \Phi(I - L \otimes_s L)^{\mathsf{T}}$. Combining with the above identity:

$$\Phi(I - L \otimes_s L)^{\mathsf{T}}(q - \widehat{q}) = P_\Phi(\Xi - \Psi_+)\widehat{q}\,.$$

Because $\Phi$ has full column rank, this identity implies that:

$$(I - L \otimes_s L)^{\mathsf{T}}(q - \widehat{q}) = (\Phi^{\mathsf{T}}\Phi)^{-1}\Phi^{\mathsf{T}}(\Xi - \Psi_+)\widehat{q}\,.$$

Using the inequlities:

$$\|(I - L \otimes_s L)^{\mathsf{T}}(q - \widehat{q})\| \geq \sigma_{\min}((I - L \otimes_s L))\|q - \widehat{q}\|\,,$$
$$(\Phi^{\mathsf{T}}\Phi)^{-1}\Phi^{\mathsf{T}}(\Xi - \Psi_+)\widehat{q} \leq \frac{\|(\Phi^{\mathsf{T}}\Phi)^{-1/2}\Phi^{\mathsf{T}}(\Xi - \Psi_+)\widehat{q}\|}{\lambda_{\min}((\Phi^{\mathsf{T}}\Phi)^{-1/2})} = \frac{\|(\Phi^{\mathsf{T}}\Phi)^{-1/2}\Phi^{\mathsf{T}}(\Xi - \Psi_+)\widehat{q}\|}{\sigma_{\min}(\Phi)}\,,$$

we obtain:

$$\|q - \widehat{q}\| \leq \frac{\|(\Phi^{\mathsf{T}}\Phi)^{-1/2}\Phi^{\mathsf{T}}(\Xi - \Psi_+)\widehat{q}\|}{\sigma_{\min}(\Phi)\sigma_{\min}(I - L \otimes_s L)}\,.$$

Next, let $\Delta = q - \widehat{q}$. By triangle inequality:

$$\|\Delta\| \leq \frac{\|(\Phi^{\mathsf{T}}\Phi)^{-1/2}\Phi^{\mathsf{T}}(\Xi - \Psi_+)\|\|\Delta\|}{\sigma_{\min}(\Phi)\sigma_{\min}(I - L \otimes_s L)} + \frac{\|(\Phi^{\mathsf{T}}\Phi)^{-1/2}\Phi^{\mathsf{T}}(\Xi - \Psi_+)q\|}{\sigma_{\min}(\Phi)\sigma_{\min}(I - L \otimes_s L)}\,.$$

The claim now follows. $\qquad\qquad\qquad\qquad\qquad\qquad\qquad\qquad\qquad\qquad\qquad\qquad\qquad\qquad\square$

In order to apply Lemma A.1, we first bound the minimum singular value $\sigma_{\min}(\Phi)$. We do this using the small-ball argument of Simchowitz et al. [33].

**Definition 2** (Definition 2.1, [33]). *Let $\{Z_t\}$ be a real-valued stochastic process that is adapted to $\{\mathcal{F}_t\}$. The process $\{Z_t\}$ satisfies the $(k, \nu, p)$ block martingale small-ball (BMSB) condition if for any $j \geq 0$ we have that:*

$$\frac{1}{k}\sum_{i=1}^{k} \mathbb{P}(|Z_{j+i}| \geq \nu|\mathcal{F}_j) \geq p \; a.s.$$

With the block martingale small-ball definition in place, we now show that the process $\langle \phi_t, y \rangle$ satisfies this condition for any fixed unit vector $y$.

**Proposition A.2.** *Given an arbitrary vector $y \in \mathcal{S}^{n+d-1}$, define the process $Z_t := \langle \phi_t, y \rangle$, the filtration $\mathcal{F}_t := \sigma(\{u_i, w_{i-1}\}_{i=0}^t)$, and matrix $C := \begin{bmatrix} I & 0 \\ K_{\text{play}} & I \end{bmatrix} \begin{bmatrix} \sigma_w I & 0 \\ 0 & \sigma_\eta I \end{bmatrix}$. Then $(Z_t)_{t\geq 1}$ satisfies the $(1, \sigma_{\min}^2(C), 1/324)$ block martingale small-ball (BMSB) condition from Definition 2. That is, almost surely, we have:*

$$\mathbb{P}(|Z_{t+1}| \geq \sigma_{\min}^2(C)|\mathcal{F}_t) \geq 1/324.$$

*Proof.* Let $Y := \text{smat}(y)$ and $\mu_t := Ax_t + Bu_t$. We have that:

$$\begin{bmatrix} x_{t+1} \\ u_{t+1} \end{bmatrix} = \begin{bmatrix} I \\ K_{\text{play}} \end{bmatrix} \mu_t + \begin{bmatrix} I & 0 \\ K_{\text{play}} & I \end{bmatrix} \begin{bmatrix} w_t \\ \eta_{t+1} \end{bmatrix}\,.$$

Therefore:

$$\langle \phi_{t+1}, y \rangle = \begin{bmatrix} x_{t+1} \\ u_{t+1} \end{bmatrix}^{\mathsf{T}} Y \begin{bmatrix} x_{t+1} \\ u_{t+1} \end{bmatrix}$$

$$= \left( \begin{bmatrix} I \\ K_{\text{play}} \end{bmatrix} \mu_t + \begin{bmatrix} I & 0 \\ K_{\text{play}} & I \end{bmatrix} \begin{bmatrix} w_t \\ \eta_{t+1} \end{bmatrix} \right)^{\mathsf{T}} Y \left( \begin{bmatrix} I \\ K_{\text{play}} \end{bmatrix} \mu_t + \begin{bmatrix} I & 0 \\ K_{\text{play}} & I \end{bmatrix} \begin{bmatrix} w_t \\ \eta_{t+1} \end{bmatrix} \right) ,$$

which is clearly a Gaussian polynomial of degree 2 given $\mathcal{F}_t$. Hence by Gaussian hyper-contractivity results (see e.g. [? ]), we have that almost surely:

$$\mathbb{E}[|Z_{t+1}|^4 | \mathcal{F}_t] \leq 81 \mathbb{E}[|Z_{t+1}|^2 | \mathcal{F}_t]^2.$$

Hence we can invoke the Paley-Zygmund inequality to conclude that for any $\theta \in (0,1)$, almost surely we have:

$$\mathbb{P}(|Z_{t+1}| \geq \sqrt{\theta \mathbb{E}[|Z_{t+1}|^2 | \mathcal{F}_t]} | \mathcal{F}_t) \geq (1-\theta)^2 \frac{\mathbb{E}[|Z_{t+1}|^2 | \mathcal{F}_t]^2}{\mathbb{E}[|Z_{t+1}|^4 | \mathcal{F}_t]} \geq \frac{(1-\theta)^2}{81}.$$

We now state an useful proposition.

**Proposition A.3.** *Let $\mu, C, Y$ be fixed and $g \sim \mathcal{N}(0, I)$. We have that:*

$$\mathbb{E}[((\mu + Cg)^{\mathsf{T}} Y (\mu + Cg))^2] \geq 2 \|C^{\mathsf{T}} Y C\|_F^2 .$$

*Proof.* Let $Z := (\mu + Cg)^{\mathsf{T}} Y (\mu + Cg)$. We know that $\mathbb{E}[Z^2] \geq \mathbb{E}[(Z - \mathbb{E}[Z])^2]$. A quick computation yields that $\mathbb{E}[Z] = \mu^{\mathsf{T}} Y \mu + \text{tr}(C^{\mathsf{T}} Y C)$. Hence

$$Z - \mathbb{E}[Z] = g^{\mathsf{T}} C^{\mathsf{T}} Y C g - \text{tr}(C^{\mathsf{T}} Y C) + 2\mu^{\mathsf{T}} Y C g .$$

Therefore,

$$\mathbb{E}[(Z - \mathbb{E}[Z])^2] \geq \mathbb{E}[(g^{\mathsf{T}} C^{\mathsf{T}} Y C g - \text{tr}(C^{\mathsf{T}} Y C))^2] = 2\|C^{\mathsf{T}} Y C\|_F^2 .$$

$\square$

Invoking Proposition A.3 and using basic properties of the Kronecker product, we have that:

$$\mathbb{E}[Z_{t+1}^2 | \mathcal{F}_t] \geq 2\|C^{\mathsf{T}} Y C\|_F^2 = 2\|(C^{\mathsf{T}} \otimes C^{\mathsf{T}}) y\|^2 \geq 2\sigma_{\min}^2 (C^{\mathsf{T}} \otimes C^{\mathsf{T}}) = 2\sigma_{\min}^4(C) .$$

The claim now follows by setting $\theta = 1/2$.

$\square$

With the BMSB bound in place, we can now utilize Proposition 2.5 of Simchowitz et al. [33] to obtain the following lower bound on the minimum singular value $\sigma_{\min}(\Phi)$.

**Proposition A.4.** *Fix $\delta \in (0,1)$. Suppose that $\sigma_\eta \leq \sigma_w$, and that $T$ exceeds:*

$$T \geq 324^2 \cdot 8 \left( (n+d)^2 \log \left( 1 + \frac{20736\sqrt{3}}{\sqrt{\delta}} \frac{(1 + \|K_{\text{play}}\|^2)^2 (\tau^2 \rho^2 n \|\Sigma_0\| + \text{tr}(P_\infty))}{\sigma_\eta^2} \right) + \log(2/\delta) \right) .$$
(A.2)

*Suppose also that $A + B K_{\text{play}}$ is $(\tau, \rho)$-stable. Then we have with probability at least $1 - \delta$,*

$$\sigma_{\min}(\Phi) \geq \frac{\sigma_\eta^2}{1296\sqrt{8}} \frac{1}{1 + \|K_{\text{play}}\|^2} \sqrt{T} .$$

*We also have with probability at least $1 - \delta$,*

$$\|\Phi^{\mathsf{T}} \Phi\| \leq \frac{12T}{\delta} (1 + \|K_{\text{play}}\|^2)^2 (\tau^2 \rho^2 n \|\Sigma_0\| + \text{tr}(P_\infty))^2 .$$

*Proof.* We first compute a crude upper bound on $\|\Phi\|$ using Markov's inequality:

$$\mathbb{P}(\|\Phi\|^2 \geq t^2) = \frac{\mathbb{E}[\lambda_{\max}(\Phi^\mathsf{T}\Phi)]}{t^2} \leq \frac{\operatorname{tr}(\mathbb{E}[\Phi^\mathsf{T}\Phi])}{t^2} \,.$$

Now we upper bound $\mathbb{E}[\|\phi_t\|^2]$. Letting $z_t = (x_t, u_t)$, we have that $\mathbb{E}[\|\phi_t\|^2] = \mathbb{E}[\|z_t\|^4] \leq 3(\mathbb{E}[\|z_t\|^2])^2$. We now bound $\mathbb{E}[\|z_t\|^2] \leq (1 + \|K_{\text{play}}\|^2)\operatorname{tr}(\Sigma_t) + \sigma_\eta^2 d$, and therefore:

$$\begin{aligned}
\sqrt{\mathbb{E}[\|\phi_t\|^2]} &\leq \sqrt{3}((1 + \|K_{\text{play}}\|^2)\operatorname{tr}(\Sigma_t) + \sigma_\eta^2 d) \\
&\leq \sqrt{3}((1 + \|K_{\text{play}}\|^2)(\tau^2 \rho^2 n \|\Sigma_0\| + \operatorname{tr}(P_\infty)) + \sigma_\eta^2 d) \\
&\leq 2\sqrt{3}(1 + \|K_{\text{play}}\|^2)(\tau^2 \rho^2 n \|\Sigma_0\| + \operatorname{tr}(P_\infty)) \,.
\end{aligned}$$

Above, the last inequality holds because $\sigma_\eta^2 d \leq \sigma_w^2 n \leq \operatorname{tr}(P_\infty)$. Therefore, we have from Markov's inequality:

$$\mathbb{P}\left(\|\Phi\| \geq \frac{\sqrt{T}}{\sqrt{\delta}} 2\sqrt{3}(1 + \|K_{\text{play}}\|^2)(\tau^2 \rho^2 n \|\Sigma_0\| + \operatorname{tr}(P_\infty))\right) \leq \delta \,.$$

Fix an $\varepsilon > 0$, and let $\mathcal{N}(\varepsilon)$ denote an $\varepsilon$-net of the unit sphere $\mathcal{S}^{(n+d)(n+d+1)/2-1}$. Next, by Proposition 2.5 of Simchowitz et al. [33] and a union bound over $\mathcal{N}(\varepsilon)$:

$$\mathbb{P}\left(\min_{v \in \mathcal{N}(\varepsilon)} \|\Phi v\| \geq \frac{\sigma_{\min}^2(C)}{324\sqrt{8}}\sqrt{T}\right) \geq 1 - (1 + 2/\varepsilon)^{(n+d)^2} e^{-\frac{T}{324^2 \cdot 8}} \,.$$

Now set

$$\varepsilon = \frac{\sqrt{\delta}}{5184\sqrt{3}} \frac{\sigma_{\min}^2(C)}{(1 + \|K_{\text{play}}\|^2)(\tau^2 \rho^2 n \|\Sigma_0\| + \operatorname{tr}(P_\infty))} \,,$$

and observe that as long as $T$ exceeds:

$$T \geq 324^2 \cdot 8 \left((n + d)^2 \log\left(1 + \frac{10368\sqrt{3}}{\sqrt{\delta}} \frac{(1 + \|K_{\text{play}}\|^2)(\tau^2 \rho^2 n \|\Sigma_0\| + \operatorname{tr}(P_\infty))}{\sigma_{\min}^2(C)}\right) + \log(2/\delta)\right) \,,$$

we have that $\mathbb{P}\left(\min_{v \in \mathcal{N}(\varepsilon)} \|\Phi v\| \geq \frac{\sigma_{\min}^2(C)}{324\sqrt{8}}\sqrt{T}\right) \geq 1 - \delta/2$. To conclude, observe that:

$$\sigma_{\min}(\Phi) = \inf_{\|v\|=1} \|\Phi v\| \geq \min_{v \in \mathcal{N}(\varepsilon)} \|\Phi v\| - \|\Phi\|\varepsilon \,,$$

and union bound over the two events. To conclude the proof, note that Lemma F.6 in Dean et al. [13] yields that $\sigma_{\min}^2(C) \geq \frac{\sigma_\eta^2}{2} \frac{1}{1 + \|K_{\text{play}}\|^2}$ since $\sigma_\eta \leq \sigma_w$. $\qquad\square$

We now turn our attention to upper bounding the self-normalized martingale terms:

$$\|(\Phi^\mathsf{T}\Phi)^{-1}\Phi^\mathsf{T}(\Xi - \Psi_+)\| \text{ and } \|(\Phi^\mathsf{T}\Phi)^{-1}\Phi^\mathsf{T}(\Xi - \Psi_+)q\| \,.$$

Our main tool here will be the self-normalized tail bounds of Abbasi-Yadkori et al. [2].

**Lemma A.5** (Corollary 1, [2]). *Let $\{\mathcal{F}_t\}$ be a filtration. Let $\{x_t\}$ be a $\mathbb{R}^{d_1}$ process that is adapted to $\{\mathcal{F}_t\}$ and let $\{w_t\}$ be a $\mathbb{R}^{d_2}$ martingale difference sequence that is adapted to $\{\mathcal{F}_t\}$. Let $V$ be a fixed positive definite $d_1 \times d_1$ matrix and define:*

$$\bar{V}_t = V + \sum_{s=1}^t x_s x_s^\mathsf{T}, \quad S_t = \sum_{s=1}^t x_s w_{s+1}^\mathsf{T} \,.$$

*(a) Suppose for any fixed unit $h \in \mathbb{R}^{d_2}$ we have that $\langle w_t, h\rangle$ is conditionally $R$-sub-Gaussian, that is:*

$$\forall \lambda \in \mathbb{R}, t \geq 1, \quad \mathbb{E}[e^{\lambda\langle w_{t+1}, h\rangle}|\mathcal{F}_t] \leq e^{\frac{\lambda^2 R^2}{2}} \,.$$

*We have that with probability at least $1 - \delta$, for all $t \geq 1$,*

$$\|\bar{V}_t^{-1/2} S_t\|^2 \leq 8R^2\left(d_2 \log 5 + \log\left(\frac{\det(\bar{V}_t)^{1/2}\det(V)^{-1/2}}{\delta}\right)\right) \,.$$

*(b) Now suppose that $\bar{\delta}$ satisfies the condition:*

$$\sum_{s=2}^{T+1} \mathbb{P}(\|w_s\| > R) \leq \bar{\delta} .$$

*Then with probability at least $1 - \delta - \bar{\delta}$, for all $1 \leq t \leq T$,*

$$\|\bar{V}_t^{-1/2} S_t\|^2 \leq 32R^2 \left( d_2 \log 5 + \log \left( \frac{\det(\bar{V}_t)^{1/2} \det(V)^{-1/2}}{\delta} \right) \right) .$$

*Proof.* Fix a unit $h \in \mathbb{R}^{d_2}$. By Corollary 1 of Abbasi-Yadkori et al. [2], we have with probability at least $1 - \delta$,

$$\|\bar{V}_t^{-1/2} S_t h\|^2 \leq 2R^2 \log \left( \frac{\det(\bar{V}_t)^{1/2} \det(V)^{-1/2}}{\delta} \right) , \quad 1 \leq t \leq T .$$

A standard covering argument yields that:

$$\|\bar{V}_t^{-1/2} S_t\|^2 \leq 4 \max_{h \in \mathcal{N}(1/2)} \|\bar{V}_t^{-1/2} S_t h\|^2 .$$

Union bounding over $\mathcal{N}(1/2)$, we obtain that:

$$\|\bar{V}_t^{-1/2} S_t\|^2 \leq 8R^2 \log \left( 5^{d_2} \frac{\det(\bar{V}_t)^{1/2} \det(V)^{-1/2}}{\delta} \right)$$
$$= 8R^2 \left( d_2 \log 5 + \log \left( \frac{\det(\bar{V}_t)^{1/2} \det(V)^{-1/2}}{\delta} \right) \right) .$$

This yields (a).

For (b), we use a simple stopping time argument to handle truncation. Define the stopping time $\tau := \inf\{t \geq 1 : \|w_t\| > R\}$ and the truncated process $\tilde{w}_t := w_t \mathbf{1}_{\tau \geq t}$. Because $\tau$ is a stopping time, this truncated process $\{\tilde{w}_t\}$ remains a martingale difference sequence. Define $Z_t = \sum_{s=1}^{t} x_s \tilde{w}_{s+1}^{\mathsf{T}}$. For any $\ell > 0$ we observe that:

$$\mathbb{P}(\exists 1 \leq t \leq T : \|\bar{V}_t^{-1/2} S_t\| > \ell)$$
$$\leq \mathbb{P}(\{\exists 1 \leq t \leq T : \|\bar{V}_t^{-1/2} S_t\| > \ell\} \cap \{\tau > T + 1\}) + \mathbb{P}(\tau \leq T + 1)$$
$$= \mathbb{P}(\{\exists 1 \leq t \leq T : \|\bar{V}_t^{-1/2} Z_t\| > \ell\} \cap \{\tau > T + 1\}) + \mathbb{P}(\tau \leq T + 1)$$
$$\leq \mathbb{P}(\exists t \geq 1 : \|\bar{V}_t^{-1/2} Z_t\| > \ell) + \mathbb{P}(\tau \leq T + 1)$$
$$\leq \mathbb{P}(\exists t \geq 1 : \|\bar{V}_t^{-1/2} Z_t\| > \ell) + \sum_{s=2}^{T+1} \mathbb{P}(\|w_s\| > R)$$
$$\leq \mathbb{P}(\exists t \geq 1 : \|\bar{V}_t^{-1/2} Z_t\| > \ell) + \bar{\delta} .$$

Now set $\ell = 32R^2 \left( d_2 \log 5 + \log \left( \frac{\det(\bar{V}_t)^{1/2} \det(V)^{-1/2}}{\delta} \right) \right)$ and using the fact that a $R$ bounded random variable is $2R$-sub-Gaussian, the claim now follows by another application of Corollary 1 from [2]. $\square$

With Lemma A.5 in place, we are ready to bound the martingale difference terms.

**Proposition A.6.** *Suppose the hypothesis of Proposition A.4 hold. With probability at least $1 - \delta$,*

$$\|(\Phi^{\mathsf{T}}\Phi)^{-1/2}\Phi^{\mathsf{T}}(\Xi - \Psi_+)q\| \leq (n+d)\sigma_w \sqrt{\tau^2 \rho^4 \|\Sigma_0\| + \|P_\infty\| + \sigma_\eta^2 \|B\|^2} (1 + \|K_{\text{eval}}\|^2)\|Q\|_F$$
$$\times \text{polylog}(n, \tau, \|\Sigma_0\|, \|P_\infty\|, \|K_{\text{play}}\|, T/\delta, 1/\sigma_\eta) ,$$
$$\|(\Phi^{\mathsf{T}}\Phi)^{-1/2}\Phi^{\mathsf{T}}(\Xi - \Psi_+)\| \leq (n+d)^2 \sigma_w \sqrt{\tau^2 \rho^4 \|\Sigma_0\| + \|P_\infty\| + \sigma_\eta^2 \|B\|^2} (1 + \|K_{\text{eval}}\|^2)$$
$$\times \text{polylog}(n, \tau, \|\Sigma_0\|, \|P_\infty\|, \|K_{\text{play}}\|, T/\delta, 1/\sigma_\eta) .$$

*Proof.* For the proof, constants $c, c_i$ will denote universal constants. Define two matrices:

$$V_1 := c_1 \frac{\sigma_\eta^4}{(1 + \|K_{\text{play}}\|^2)^2} T \cdot I \, ,$$

$$V_2 := c_2 \frac{T}{\delta} (1 + \|K_{\text{play}}\|^2)^2 (\tau^2 \rho^2 n \|\Sigma_0\| + \text{tr}(P_\infty))^2 \cdot I \, .$$

By Proposition A.4, with probability at least $1 - \delta/2$, we have that:

$$V_1 \preceq \Phi^\mathsf{T} \Phi \preceq V_2 \, .$$

Call this event $\mathcal{E}_1$.

Next, we have:

$$
\begin{aligned}
&\mathbb{E}[x_{t+1} x_{t+1}^\mathsf{T} | x_t, u_t] - x_{t+1} x_{t+1}^\mathsf{T} \\
&= \mathbb{E}[(Ax_t + Bu_t + w_t)(Ax_t + Bu_t + w_t)^\mathsf{T} | x_t, u_t] - (Ax_t + Bu_t + w_t)(Ax_t + Bu_t + w_t)^\mathsf{T} \\
&= (Ax_t + Bu_t)(Ax_t + Bu_t)^\mathsf{T} + \sigma_w^2 I \\
&\quad - (Ax_t + Bu_t)(Ax_t + Bu_t)^\mathsf{T} - (Ax_t + Bu_t)w_t^\mathsf{T} - w_t(Ax_t + Bu_t)^\mathsf{T} - w_t w_t^\mathsf{T} \\
&= \sigma_w^2 I - w_t w_t^\mathsf{T} - (Ax_t + Bu_t)w_t^\mathsf{T} - w_t(Ax_t + Bu_t)^\mathsf{T} \, .
\end{aligned}
$$

Therefore,

$$
\begin{aligned}
&\mathbb{E}[\psi_{t+1} | x_t, u_t] - \psi_{t+1} \\
&= \text{svec}\left( \begin{bmatrix} I \\ K_{\text{eval}} \end{bmatrix} (\sigma_w^2 I - w_t w_t^\mathsf{T} - (Ax_t + Bu_t)w_t^\mathsf{T} - w_t(Ax_t + Bu_t)^\mathsf{T}) \begin{bmatrix} I \\ K_{\text{eval}} \end{bmatrix}^\mathsf{T} \right) \, .
\end{aligned}
$$

Taking the inner product of this term with $q$,

$$
\begin{aligned}
&(\mathbb{E}[\psi_{t+1} | x_t, u_t] - \psi_{t+1})^\mathsf{T} q \\
&= \text{tr}\left( (\sigma_w^2 I - w_t w_t^\mathsf{T} - (Ax_t + Bu_t)w_t^\mathsf{T} - w_t(Ax_t + Bu_t)^\mathsf{T}) \begin{bmatrix} I \\ K_{\text{eval}} \end{bmatrix}^\mathsf{T} Q \begin{bmatrix} I \\ K_{\text{eval}} \end{bmatrix} \right) \\
&= \text{tr}\left( (\sigma_w^2 I - w_t w_t^\mathsf{T}) \begin{bmatrix} I \\ K_{\text{eval}} \end{bmatrix}^\mathsf{T} Q \begin{bmatrix} I \\ K_{\text{eval}} \end{bmatrix} \right) - 2 w_t^\mathsf{T} \begin{bmatrix} I \\ K_{\text{eval}} \end{bmatrix}^\mathsf{T} Q \begin{bmatrix} I \\ K_{\text{eval}} \end{bmatrix} (Ax_t + Bu_t) \, .
\end{aligned}
$$

By the Hanson-Wright inequality (see e.g. **?** ]), with probability at least $1 - \delta/T$,

$$\left| \text{tr}\left( (\sigma_w^2 I - w_t w_t^\mathsf{T}) \begin{bmatrix} I \\ K_{\text{eval}} \end{bmatrix}^\mathsf{T} Q \begin{bmatrix} I \\ K_{\text{eval}} \end{bmatrix} \right) \right| \leq c_1 \sigma_w^2 (1 + \|K_{\text{eval}}\|^2) \|Q\|_F \log(T/\delta) \, .$$

Now, let $L_{\text{play}} := A + B K_{\text{play}}$. By Proposition 4.7 in Tu and Recht [34], with probability at least $1 - \delta/T$,

$$
\begin{aligned}
&\left| w_t^\mathsf{T} \begin{bmatrix} I \\ K_{\text{eval}} \end{bmatrix}^\mathsf{T} Q \begin{bmatrix} I \\ K_{\text{eval}} \end{bmatrix} (Ax_t + Bu_t) \right| \\
&\leq c_1 \sigma_w (1 + \|K_{\text{eval}}\|^2) \sqrt{\|L_{\text{play}}^{t+1} \Sigma_0 (L_{\text{play}}^{t+1})^\mathsf{T}\| + \|L_{\text{play}} P_t L_{\text{play}}^\mathsf{T}\| + \sigma_\eta^2 \|B\|^2} \|Q\|_F \log(T/\delta) \\
&\leq c_1 \sigma_w (1 + \|K_{\text{eval}}\|^2) \sqrt{\tau^2 \rho^{2(t+1)} \|\Sigma_0\| + \|P_\infty\| + \sigma_\eta^2 \|B\|^2} \|Q\|_F \log(T/\delta) \, ,
\end{aligned}
$$

where the inequality above comes from $P_t \preceq P_\infty$ and $L_{\text{play}} P_\infty L_{\text{play}}^\mathsf{T} \preceq P_\infty$. Therefore, we have:

$$
\begin{aligned}
&|(\mathbb{E}[\psi_{t+1} | x_t, u_t] - \psi_{t+1})^\mathsf{T} v| \\
&\leq c_2 (\sigma_w^2 + \sigma_w \sqrt{\tau^2 \rho^{2(t+1)} \|\Sigma_0\| + \|P_\infty\| + \sigma_\eta^2 \|B\|^2}) (1 + \|K_{\text{eval}}\|^2) \|Q\|_F \log(T/\delta) \\
&\leq c_3 \sigma_w \sqrt{\tau^2 \rho^{2(t+1)} \|\Sigma_0\| + \|P_\infty\| + \sigma_\eta^2 \|B\|^2} (1 + \|K_{\text{eval}}\|^2) \|Q\|_F \log(T/\delta) \, .
\end{aligned}
$$

The last inequality holds because $P_\infty \succeq \sigma_w^2 I$ and hence $\sigma_w \leq \|P_\infty\|^{1/2}$. Therefore we can set

$$R = c_3 \sigma_w \sqrt{\tau^2 \rho^4 \|\Sigma_0\| + \|P_\infty\| + \sigma_\eta^2 \|B\|^2} (1 + \|K_{\text{eval}}\|^2) \|Q\|_F \log(T/\delta)\,,$$

and invoke Lemma A.5 to conclude that with probability at least $1 - \delta/2$,

$$\|(V_1 + \Phi^\mathsf{T}\Phi)^{-1/2}\Phi^\mathsf{T}(\Xi - \Psi_+)v\| \leq c_4(n+d)R + c_5 R\sqrt{\log(\det((V_1 + \Phi^\mathsf{T}\Phi)V_1^{-1})^{1/2}/\delta)}\,.$$

Call this event $\mathcal{E}_2$.

For the remainder of the proof we work on $\mathcal{E}_1 \cap \mathcal{E}_2$, which has probability at least $1 - \delta$. Since $\Phi^\mathsf{T}\Phi \succeq V_1$, we have that $(\Phi^\mathsf{T}\Phi)^{-1} \leq 2(V_1 + \Phi^\mathsf{T}\Phi)^{-1}$. Therefore, by another application of Lemma A.5:

$$\|(\Phi^\mathsf{T}\Phi)^{-1/2}\Phi^\mathsf{T}(\Xi - \Psi_+)\|$$
$$\leq \sqrt{2}\|(V_1 + \Phi^\mathsf{T}\Phi)^{-1/2}\Phi^\mathsf{T}(\Xi - \Psi_+)\|$$
$$\leq c_6(n+d)R + c_7 R\sqrt{\log(\det((V_1 + \Phi^\mathsf{T}\Phi)V_1^{-1})^{1/2}/\delta)}$$
$$\leq c_6(n+d)R + c_7 R\sqrt{\log(\det((V_1 + V_2)V_1^{-1})^{1/2}/\delta)}$$
$$\leq c_6(n+d)R + c_8 R(n+d)\sqrt{\log\left(\frac{(1 + \|K_{\text{play}}\|^2)^4}{\delta} \frac{(\tau^2 \rho^2 n \|\Sigma_0\| + \text{tr}(P_\infty))^2}{\sigma_\eta^4}\right)}$$
$$\leq c(n+d)R\,\text{polylog}(n, \tau, \|\Sigma_0\|, \|P_\infty\|, \|K_{\text{play}}\|, 1/\delta, 1/\sigma_\eta)\,.$$

Next, we bound:

$$\|\mathbb{E}[\psi_{t+1}|x_t, u_t] - \psi_{t+1}\|$$
$$\leq \left\|\begin{bmatrix} I \\ K_{\text{eval}} \end{bmatrix}(\sigma_w^2 I - w_t w_t^\mathsf{T})\begin{bmatrix} I \\ K_{\text{eval}} \end{bmatrix}^\mathsf{T}\right\|_F + \left\|\begin{bmatrix} I \\ K_{\text{eval}} \end{bmatrix} w_t (Ax_t + Bu_t)^\mathsf{T} \begin{bmatrix} I \\ K_{\text{eval}} \end{bmatrix}^\mathsf{T}\right\|_F$$
$$\leq (1 + \|K_{\text{eval}}\|^2)(\|\sigma_w^2 I - w_t w_t^\mathsf{T}\|_F + \|w_t(Ax_t + Bu_t)^\mathsf{T}\|_F)\,.$$

Now, by standard Gaussian concentration results, with probability $1 - \delta/T$,

$$\|\sigma_w^2 I - w_t w_t^\mathsf{T}\|_F \leq c\sigma_w^2(n + \log(T/\delta))\,,$$

and also

$$\|w_t(Ax_t + Bu_t)^\mathsf{T}\|_F$$
$$\leq c\sigma_w(\sqrt{n} + \sqrt{\log(T/\delta)})\left(\sqrt{\text{tr}(L_{\text{play}}^{t+1}\Sigma_0(L_{\text{play}}^{t+1})^\mathsf{T}) + \text{tr}(L_{\text{play}}P_t L_{\text{play}}^\mathsf{T}) + \sigma_\eta^2\|B\|_F^2}\right.$$
$$\left. + \sqrt{\|L_{\text{play}}^{t+1}\Sigma_0(L_{\text{play}}^{t+1})^\mathsf{T}\| + \|L_{\text{play}}P_t L_{\text{play}}^\mathsf{T}\| + \sigma_\eta^2\|B\|^2}\sqrt{\log(T/\delta)}\right)$$
$$\leq c\sigma_w(n+d)\sqrt{\tau^2 \rho^4 \|\Sigma_0\| + \|P_\infty\| + \sigma_\eta^2\|B\|}\log(T/\delta)\,.$$

Therefore, with probability $1 - \delta/T$,

$$\|\mathbb{E}[\psi_{t+1}|x_t, u_t] - \psi_{t+1}\|$$
$$\leq c(1 + \|K_{\text{eval}}\|^2)(n+d)\sigma_w\sqrt{\tau^2\rho^4\|\Sigma_0\| + \|P_\infty\| + \sigma_\eta^2\|B\|^2}\log(T/\delta)\,.$$

$\square$

We are now in a position to prove Theorem 2.1. We first observe that we can lower bound $\sigma_{\min}(I - L \otimes_s L)$ using the $(\tau, \rho)$-stability of $A + BK_{\text{eval}}$. This is because for $k \geq 1$,

$$\|L^k\| = \left\|\begin{bmatrix} I \\ K_{\text{eval}} \end{bmatrix}(A + BK_{\text{eval}})^{k-1}\begin{bmatrix} A & B \end{bmatrix}\right\|$$
$$\leq 2\|K_{\text{eval}}\|_+ \|\begin{bmatrix} A & B \end{bmatrix}\|\tau\rho^{k-1}$$
$$\leq \frac{2\|K_{\text{eval}}\|_+ \max\{1, \sqrt{\|A\|^2 + \|B\|^2}\}}{\rho}\tau \cdot \rho^k\,.$$

Hence we see that $L$ is $(\frac{2\|K_{\mathrm{eval}}\|_+ \max\{1, \sqrt{\|A\|^2 + \|B\|^2}\}}{\rho}\tau, \rho)$-stable. Next, we know that $\sigma_{\min}(I - L \otimes_s L) = \frac{1}{\|(I - L \otimes_s L)^{-1}\|}$. Therefore, for any unit norm $v$,

$$\|(I - L \otimes_s L)^{-1} v\| = \|(I - L \otimes_s L)^{-1} \mathrm{svec}(\mathrm{smat}(v))\| = \|\mathsf{dlyap}(L^{\mathsf{T}}, \mathrm{smat}(v))\|_F$$
$$\leq \frac{4\|K_{\mathrm{eval}}\|_+^2 (\|A\|^2 + \|B\|^2)_+ \tau^2}{\rho^2(1 - \rho^2)} \ .$$

Here, the last inequality uses Proposition E.5. Hence we have the bound:

$$\sigma_{\min}(I - L \otimes_s L) \geq \frac{\rho^2(1 - \rho^2)}{4\|K_{\mathrm{eval}}\|_+^2 (\|A\|^2 + \|B\|^2)_+ \tau^2} \ .$$

By Proposition A.4, as long as $T \geq \widetilde{O}(1)(n + d)^2$ with probability at least $1 - \delta/2$:

$$\sigma_{\min}(\Phi) \geq c \frac{\sigma_\eta^2}{\|K_{\mathrm{play}}\|_+^2} \sqrt{T} \ .$$

By Proposition A.6, with probability at least $1 - \delta/2$:

$$\|(\Phi^{\mathsf{T}}\Phi)^{-1/2}\Phi^{\mathsf{T}}(\Xi - \Psi_+)q\| \leq (n + d)\sigma_w \sqrt{\tau^2 \rho^4 \|\Sigma_0\| + \|P_\infty\| + \sigma_\eta^2 \|B\|^2} \|K_{\mathrm{eval}}\|_+^2 \|Q^{K_{\mathrm{eval}}}\|_F \widetilde{O}(1) \ ,$$

$$\|(\Phi^{\mathsf{T}}\Phi)^{-1/2}\Phi^{\mathsf{T}}(\Xi - \Psi_+)\| \leq (n + d)^2 \sigma_w \sqrt{\tau^2 \rho^4 \|\Sigma_0\| + \|P_\infty\| + \sigma_\eta^2 \|B\|^2} \|K_{\mathrm{eval}}\|_+^2 \widetilde{O}(1) \ .$$

We first check the condition

$$\frac{\|(\Phi^{\mathsf{T}}\Phi)^{-1/2}\Phi^{\mathsf{T}}(\Xi - \Psi_+)\|}{\sigma_{\min}(\Phi)\sigma_{\min}(I - L \otimes_s L)} \leq 1/2 \ ,$$

from Lemma A.1. A sufficient condition is that $T$ satisfies:

$$T \geq \widetilde{O}(1)\frac{\|K_{\mathrm{play}}\|_+^4}{\sigma_\eta^4} \cdot (n + d)^4 \sigma_w^2 (\tau^2 \rho^4 \|\Sigma_0\| + \|P_\infty\| + \sigma_\eta^2 \|B\|^2)$$

$$\times \|K_{\mathrm{eval}}\|_+^4 \cdot \frac{\|K_{\mathrm{eval}}\|_+^4 (\|A\|^2 + \|B\|^2)_+^2 \tau^4}{\rho^4(1 - \rho^2)^2}$$

$$= \widetilde{O}(1)\frac{\tau^4}{\rho^4(1 - \rho^2)^2} \frac{(n + d)^4}{\sigma_\eta^4} \sigma_w^2 (\tau^2 \rho^4 \|\Sigma_0\| + \|P_\infty\| + \sigma_\eta^2 \|B\|^2)$$

$$\times \|K_{\mathrm{play}}\|_+^4 \|K_{\mathrm{eval}}\|_+^8 (\|A\|^4 + \|B\|^4)_+ \ .$$

Once this condition on $T$ is satisfied, then we have that the error $\|\widehat{q} - q\|$ is bounded by:

$$\widetilde{O}(1)\frac{\|K_{\mathrm{play}}\|_+^2}{\sigma_\eta^2 \sqrt{T}} \cdot (n + d)\sigma_w \sqrt{\tau^2 \rho^4 \|\Sigma_0\| + \|P_\infty\| + \sigma_\eta^2 \|B\|^2}$$

$$\times \|K_{\mathrm{eval}}\|_+^2 \|Q^{K_{\mathrm{eval}}}\|_F \cdot \frac{\|K_{\mathrm{eval}}\|_+^2 (\|A\|^2 + \|B\|^2)_+ \tau^2}{\rho^2(1 - \rho^2)}$$

$$= \widetilde{O}(1)\frac{\tau^2}{\rho^2(1 - \rho^2)} \frac{(n + d)}{\sigma_\eta^2 \sqrt{T}} \sigma_w \sqrt{\tau^2 \rho^4 \|\Sigma_0\| + \|P_\infty\| + \sigma_\eta^2 \|B\|^2}$$

$$\times \|K_{\mathrm{play}}\|_+^2 \|K_{\mathrm{eval}}\|_+^4 (\|A\|^2 + \|B\|^2)_+ \|Q^{K_{\mathrm{eval}}}\|_F \ .$$

Theorem 2.1 now follows from Lemma A.1.

# B   Analysis for LSPI

In this section we study the non-asymptotic behavior of LSPI. Our analysis proceeds in two steps. We first understand the behavior of exact policy iteration on LQR. Then, we study the effects of introducing errors into the policy iteration updates.

## B.1 Exact Policy Iteration

Exactly policy iteration works as follows. We start with a stabilizing controller $K_0$ for $(A, B)$, and let $V_0$ denote its associated value function. We then apply the following recursions for $t = 0, 1, 2, ...$:

$$K_{t+1} = -(S + B^\mathsf{T} V_t B)^{-1} B^\mathsf{T} V_t A \,, \tag{B.1}$$

$$V_{t+1} = \mathsf{dlyap}(A + BK_{t+1}, S + K_{t+1}^\mathsf{T} R K_{t+1}) \,. \tag{B.2}$$

Note that this recurrence is related to, but different from, that of *value iteration*, which starts from a PSD $V_0$ and recurses:

$$V_{t+1} = A^\mathsf{T} V_t A - A^\mathsf{T} V_t B(S + B^\mathsf{T} V_t B)^{-1} B^\mathsf{T} V_t A + S \,.$$

While the behavior of value iteration for LQR is well understood (see e.g. Lincoln and Rantzer [23] or **?** ]), the behavior of policy iteration is less studied. Fazel et al. [16] show that policy iteration is equivalent to the Gauss-Newton method on the objective $J(K)$ with a specific step-size, and give a simple analysis which shows linear convergence to the optimal controller. In this section, we present an analysis of the behavior of exact policy iteration that builds on top of the fixed-point theory from Lee and Lim [22]. A key component of our analysis is the following invariant metric $\delta_\infty$ on positive definite matrices:

$$\delta_\infty(A, B) := \|\log(A^{-1/2} B A^{-1/2})\| \,.$$

Various properties of $\delta_\infty$ are reviewed in Appendix D.

Our analysis proceeds as follows. First, we note by the matrix inversion lemma:

$$S + A^\mathsf{T}(BR^{-1}B^\mathsf{T} + V^{-1})^{-1}A = S + A^\mathsf{T}VA - A^\mathsf{T}VB(R + B^\mathsf{T}VB)^{-1}B^\mathsf{T}VA =: F(V) \,.$$

Let $V_\star$ be the unique positive definite solution to $V = F(V)$. For any positive definite $V$ we have by Lemma D.2:

$$\delta_\infty(F(V), V_\star) \leq \frac{\alpha}{\lambda_{\min}(S) + \alpha} \delta_\infty(V, V_\star) \,, \tag{B.3}$$

with $\alpha = \max\{\lambda_{\max}(A^\mathsf{T}VA), \lambda_{\max}(A^\mathsf{T}V_\star A)\}$. Indeed, (B.3) gives us another method to analyze value iteration, since it shows that the Riccati operator $F(V)$ is contractive in the $\delta_\infty$ metric. Our next result combines this contraction property with the policy iteration analysis of Bertsekas [7].

**Proposition B.1** (Policy Iteration for LQR)**.** *Suppose that $S, R$ are positive definite and there exists a unique positive definite solution to the discrete algebraic Riccati equation (DARE). Let $K_0$ be a stabilizing policy for $(A, B)$ and let $V_0 = \mathsf{dlyap}(A + BK_0, S + K_0^\mathsf{T} R K_0)$. Consider the following sequence of updates for $t = 0, 1, 2, ...$:*

$$K_{t+1} = -(R + B^\mathsf{T} V_t B)^{-1} B^\mathsf{T} V_t A \,,$$

$$V_{t+1} = \mathsf{dlyap}(A + BK_{t+1}, S + K_{t+1}^\mathsf{T} R K_{t+1}) \,.$$

*The following statements hold:*

*(i) $K_t$ stabilizes $(A, B)$ for all $t = 0, 1, 2, ...$,*

*(ii) $V_\star \preceq V_{t+1} \preceq V_t$ for all $t = 0, 1, 2, ...$,*

*(iii) $\delta_\infty(V_{t+1}, V_\star) \leq \rho \cdot \delta_\infty(V_t, V_\star)$ for all $t = 0, 1, 2, ...$, with $\rho := \frac{\lambda_{\max}(A^\mathsf{T} V_0 A)}{\lambda_{\min}(S) + \lambda_{\max}(A^\mathsf{T} V_0 A)}$. Consequently, $\delta_\infty(V_t, V_\star) \leq \rho^t \cdot \delta_\infty(V_0, V_\star)$ for $t = 0, 1, 2, ....$*

*Proof.* We first prove (i) and (ii) using the argument of Proposition 1.3 from Bertsekas [7].

Let $c(x, u) = x^\mathsf{T} Sx + u^\mathsf{T} Ru$, $f(x, u) = Ax + Bu$, and $V^K(x_1) = \sum_{t=1}^{\infty} c(x_t, u_t)$ with $x_{t+1} = f(x_t, u_t)$ and $u_t = Kx_t$. Let $V_t = V^{K_t}$. With these definitions, we have that for all $x$:

$$K_{t+1}x = \arg\min_u c(x, u) + V_t(f(x, u)) \,.$$

Therefore,

$$
\begin{aligned}
V_t(x) &= c(x, K_t x) + V_t(f(x, K_t x)) \\
&\geq c(x, K_{t+1} x) + V_t(f(x, K_{t+1} x)) \\
&= c(x, K_{t+1} x) + c(f(x, K_{t+1} x), K_t f(x, K_{t+1} x)) + V_t(f(f(x, K_{t+1} x), K_t f(x, K_{t+1} x))) \\
&\geq c(x, K_{t+1} x) + c(f(x, K_{t+1} x), K_{t+1} f(x, K_{t+1} x)) + V_t(f(f(x, K_{t+1} x), K_{t+1} f(x, K_{t+1} x))) \\
&\vdots \\
&\geq V_{t+1}(x) \, .
\end{aligned}
$$

This proves (i) and (ii).

Now, observe that by partial minimization of a strongly convex quadratic:

$$
\begin{aligned}
c(x, K_{t+1} x) + V_t(f(x, K_{t+1} x)) &= \min_u c(x, u) + V_t(f(x, u)) \\
&= x^{\mathsf{T}}(S + A^{\mathsf{T}} V_t A - A^{\mathsf{T}} V_t B (R + B^{\mathsf{T}} V_t B)^{-1} B^{\mathsf{T}} V_t A) x \\
&= x^{\mathsf{T}} F(V_t) x \, .
\end{aligned}
$$

Combined with the above inequalities, this shows that $V_{t+1} \preceq F(V_t) \preceq V_t$. Therefore, by (B.3) and Proposition D.4,

$$
\begin{aligned}
\delta_\infty(V_{t+1}, V_\star) &\leq \delta_\infty(F(V_t), V_\star) \\
&= \delta_\infty(F(V_t), F(V_\star)) \\
&\leq \frac{\alpha_t}{\lambda_{\min}(Q) + \alpha_t} \delta_\infty(V_t, V_\star) \, ,
\end{aligned}
$$

where $\alpha_t = \max\{\lambda_{\max}(A^{\mathsf{T}} V_t A), \lambda_{\max}(A^{\mathsf{T}} V_\star A)\} = \lambda_{\max}(A^{\mathsf{T}} V_t A)$, since $V_\star \preceq V_t$. But since $V_t \preceq V_0$, we can upper bound $\alpha_t \leq \lambda_{\max}(A^{\mathsf{T}} V_0 A)$. This proves (iii). □

## B.2 Approximate Policy Iteration

We now turn to the analysis of approximate policy iteration. Before analyzing Algorithm 2, we analyze a slightly more general algorithm described in Algorithm 4

---

**Algorithm 4** Approximate Policy Iteration for LQR (offline)

---

**Input:** Initial stabilizing controller $K_0$, $N$ number of policy iterations, $T$ length of rollout for estimation, $\sigma_\eta^2$ exploration variance.
1: **for** $t = 0, ..., N-1$ **do**
2:      Collect a trajectory $\mathcal{D}_t = \{(x_k^{(t)}, u_k^{(t)}, x_{k+1}^{(t)})\}_{k=1}^T$ using input $u_k^{(t)} = K_0 x_k^{(t)} + \eta_k^{(t)}$, with $\eta_k^{(t)} \sim \mathcal{N}(0, \sigma_\eta^2 I)$.
3:      $\widehat{Q}_t = \mathsf{EstimateQ}(\mathcal{D}_t, K_t)$.
4:      $K_{t+1} = G(\widehat{Q}_t)$. [c.f. (2.10)]
5: **end for**
6: **return** $K_N$.

---

In Algorithm 4, the procedure $\mathsf{EstimateQ}$ takes as input an off-policy trajectory $\mathcal{D}_t$ and a policy $K_t$, and returns an estimate $\widehat{Q}_t$ of the true $Q$ function $Q_t$. We will analyze Algorithm 4 first assuming that the procedure $\mathsf{EstimateQ}$ delivers an estimate with a certain level of accuracy. In order to do this, we define the sequence of variables:

(i) $Q_t$ is true state-value function for $K_t$.

(ii) $V_t$ is true value function for $K_t$.

(iii) $\overline{K}_{t+1} = G(Q_t)$.

(iv) $\overline{V}_t$ is true value function for $\overline{K}_t$.

The following proposition is our main result regarding Algorithm 4.

**Proposition B.2.** *Consider the sequence of updates defined by Algorithm 4. Suppose we start with a stabilizing $K_0$ and let $V_0$ denote its value function. Fix an $\varepsilon > 0$. Define the following variables:*

$$\mu := \min\{\lambda_{\min}(S), \lambda_{\min}(R)\} \,,$$

$$Q_{\max} := \max\{\|S\|, \|R\|\} + 2(\|A\|^2 + \|B\|^2)\|V_0\| \,,$$

$$\gamma := \frac{2\|A\|^2\|V_0\|}{\mu + 2\|A\|^2\|V_0\|} \,,$$

$$N_0 := \lceil \frac{1}{1-\gamma} \log(2\delta_\infty(V_0, V_\star)/\varepsilon) \rceil \,,$$

$$\tau := \sqrt{\frac{2\|V_0\|}{\mu}} \,,$$

$$\rho := \sqrt{1 - 1/\tau^2} \,,$$

$$\overline{\rho} := \mathsf{Avg}(\rho, 1) \,,$$

*where $\mathsf{Avg}(x,y) = \frac{x+y}{2}$. Let $N_1 \geq N_0$. Suppose the estimates $\widehat{Q}_t$ output by $\mathsf{EstimateQ}$ satisfy, for all $0 \leq t \leq N_1 - 1$, $\widehat{Q}_t \succeq \mu I$ and furthermore,*

$$\|\widehat{Q}_t - Q_t\| \leq \min\left\{ \frac{\|V_0\|}{N_1}, \varepsilon\mu(1-\gamma) \right\} \left( \frac{\mu}{28} \frac{(1-\overline{\rho}^2)^2}{\tau^5} \frac{1}{\|B\|_+ \max\{\|S\|, \|R\|\}} \frac{\mu^3}{Q_{\max}^3} \right) \,.$$

*Then we have for any $N$ satisfying $N_0 \leq N \leq N_1$ the bound $\delta_\infty(V_N, V_\star) \leq \varepsilon$. We also have that for all $0 \leq t \leq N_1$, $A + BK_t$ is $(\tau, \overline{\rho})$-stable and $\|K_t\| \leq 2Q_{\max}/\mu$.*

*Proof.* We first start by observing that if $V, V_0$ are value functions satisfying $V \preceq V_0$, then their state-value functions also satisfy $Q \preceq Q_0$. This is because

$$Q = \begin{bmatrix} S & 0 \\ 0 & R \end{bmatrix} + \begin{bmatrix} A^\mathsf{T} \\ B^\mathsf{T} \end{bmatrix} V \begin{bmatrix} A & B \end{bmatrix}$$

$$\preceq \begin{bmatrix} S & 0 \\ 0 & R \end{bmatrix} + \begin{bmatrix} A^\mathsf{T} \\ B^\mathsf{T} \end{bmatrix} V_0 \begin{bmatrix} A & B \end{bmatrix} = Q_0 \,.$$

From this we also see that any state-value function satisfies $Q \succeq \begin{bmatrix} S & 0 \\ 0 & R \end{bmatrix}$.

The proof proceeds as follows. We observe that since $\overline{V}_{t+1} \preceq V_t$ (Proposition B.1-(ii)):

$$V_t = V_t - \overline{V}_t + \overline{V}_t - V_{t-1} + V_{t-1} \preceq V_t - \overline{V}_t + V_{t-1} \,.$$

Therefore, by triangle inequality we have $\|V_t\| \leq \|V_t - \overline{V}_t\| + \|V_{t-1}\|$. Supposing for now that we can ensure for all $1 \leq t \leq N_1$:

$$\|V_t - \overline{V}_t\| \leq \frac{\|V_0\|}{N} \,, \tag{B.4}$$

unrolling the recursion for $\|V_t\|$ for $N_1$ steps ensures that $\|V_t\| \leq 2\|V_0\|$ for all $0 \leq t \leq N_1$. Furthermore,

$$\|Q_t\| \leq \max\{\|S\|, \|R\|\} + \|[A \quad B]\|^2\|V_t\|$$

$$\leq \max\{\|S\|, \|R\|\} + 2(\|A\|^2 + \|B\|^2)\|V_0\|$$

$$= Q_{\max} \,.$$

for all $0 \leq t \leq N_1$.

Now, by triangle inequality and Proposition B.1-(iii), for all $0 \leq t \leq N_1 - 1$,

$$\delta_\infty(V_{t+1}, V_\star) \leq \delta_\infty(V_{t+1}, \overline{V}_{t+1}) + \delta_\infty(\overline{V}_{t+1}, V_\star)$$

$$\leq \delta_\infty(V_{t+1}, \overline{V}_{t+1}) + \gamma \cdot \delta_\infty(V_t, V_\star)$$

$$\leq \frac{\|V_{t+1} - \overline{V}_{t+1}\|}{\mu} + \gamma \cdot \delta_\infty(V_t, V_\star) \,, \tag{B.5}$$

where $\gamma = \frac{2\|A\|^2\|V_0\|}{\mu + 2\|A\|^2\|V_0\|}$, and the last inequality uses Proposition D.3 combined with the fact that $V_{t+1} \succeq \mu I$ and $\overline{V}_{t+1} \succeq \mu I$.

We now focus on bounding $\|V_{t+1} - \overline{V}_{t+1}\|$. To do this, we first bound $\|K_{t+1} - \overline{K}_{t+1}\|$, and then use the Lyapunov perturbation result from Section E. First, observe the simple bounds:

$$\|\overline{K}_{t+1}\| = \|G(Q_t)\| \leq \frac{\|Q_t\|}{\mu} \leq \frac{Q_{\max}}{\mu},$$

$$\|K_{t+1}\| = \|G(\widehat{Q}_t)\| \leq \frac{\|\widehat{Q}_t\|}{\mu} \leq \frac{\Delta + Q_{\max}}{\mu} \leq \frac{2Q_{\max}}{\mu}.$$

where the second bound uses the assumption that the estimates $\widehat{Q}_t$ satisfy $\widehat{Q}_t \succeq \mu I$ and $\|\widehat{Q}_t - Q_t\| \leq \Delta$ with

$$\Delta \leq Q_{\max}. \tag{B.6}$$

Now, by Proposition E.3 we have:

$$\begin{aligned}
\|K_{t+1} - \overline{K}_{t+1}\| &= \|G(\widehat{Q}_t) - G(Q_t)\| \\
&\leq \frac{(1 + \|\overline{K}_{t+1}\|)\|\widehat{Q}_t - Q_t\|}{\mu} \\
&\leq \frac{(1 + Q_{\max}/\mu)\Delta}{\mu} \\
&\leq \frac{2Q_{\max}}{\mu^2}\Delta.
\end{aligned}$$

Above, the last inequality holds since $Q_{\max} \geq \mu$ by definition.

By Proposition E.4, because $\overline{V}_{t+1} \preceq V_t$, we know that $\overline{K}_{t+1}$ satisfies for all $k \geq 0$:

$$\begin{aligned}
\|(A + B\overline{K}_{t+1})^k\| &\leq \sqrt{\frac{\|V_t\|}{\lambda_{\min}(S)}} \cdot \sqrt{1 - \lambda_{\min}(V_t^{-1}S)}^k \\
&\leq \sqrt{\frac{2\|V_0\|}{\mu}}\sqrt{1 - \frac{\mu}{2\|V_0\|}}^k = \tau \cdot \rho^k.
\end{aligned}$$

Let us now assume that $\Delta$ satisfies:

$$\frac{2Q_{\max}}{\mu^2} \cdot \Delta \leq \frac{1 - \rho}{2\tau\|B\|}. \tag{B.7}$$

Then by Lemma E.1, we know that $\|(A + BK_{t+1})^k\| \leq \tau \cdot \overline{\rho}^k$. Hence, we have that $A + BK_{t+1}$ is $(\tau, \overline{\rho})$-stable.

Next, by the Lyapunov perturbation result of Proposition E.6,

$$\begin{aligned}
&\|V_{t+1} - \overline{V}_{t+1}\| \\
&= \|\mathsf{dlyap}(A + BK_{t+1}, S + K_{t+1}^{\mathsf{T}}RK_{t+1}) - \mathsf{dlyap}(A + B\overline{K}_{t+1}, S + \overline{K}_{t+1}^{\mathsf{T}}R\overline{K}_{t+1})\| \\
&\leq \frac{\tau^2}{1 - \overline{\rho}^2}\|K_{t+1}^{\mathsf{T}}RK_{t+1} - \overline{K}_{t+1}^{\mathsf{T}}R\overline{K}_{t+1}\| \\
&\quad + \frac{\tau^4}{(1 - \overline{\rho}^2)^2}\|B(K_{t+1} - \overline{K}_{t+1})\|(\|A + BK_{t+1}\| + \|A + B\overline{K}_{t+1}\|)\|S + \overline{K}_{t+1}^{\mathsf{T}}R\overline{K}_{t+1}\|.
\end{aligned}$$

We bound:

$$\|K_{t+1}^{\mathsf{T}} R K_{t+1} - \overline{K}_{t+1}^{\mathsf{T}} R \overline{K}_{t+1}\| \le \|R\|\|K_{t+1} - \overline{K}_{t+1}\|(\|K_{t+1}\| + \|\overline{K}_{t+1}\|)$$
$$\le \frac{6\|R\|Q_{\mathrm{max}}^2}{\mu^3}\Delta \,,$$
$$\|B(K_{t+1} - \overline{K}_{t+1})\| \le \frac{2\|B\|Q_{\mathrm{max}}}{\mu^2}\Delta \,,$$
$$\max\{\|A + BK_{t+1}\|, \|A + B\overline{K}_{t+1}\|\} \le \tau \,,$$
$$\|S + \overline{K}_{t+1}^{\mathsf{T}} R \overline{K}_{t+1}\| \le \|S\| + \frac{\|R\|Q_{\mathrm{max}}^2}{\mu^2} \,.$$

Therefore,

$$
\begin{aligned}
\|V_{t+1} - \overline{V}_{t+1}\| &\le \frac{\tau^2}{1-\overline{\rho}^2}\frac{6\|R\|Q_{\mathrm{max}}^2}{\mu^3}\Delta + 8\frac{\tau^5}{(1-\overline{\rho}^2)^2}\|B\|\max\{\|S\|, \|R\|\}\frac{Q_{\mathrm{max}}^3}{\mu^4}\Delta \\
&= \frac{1}{\mu}\left(\frac{\tau^2}{1-\overline{\rho}^2}\frac{6\|R\|Q_{\mathrm{max}}^2}{\mu^2} + 8\frac{\tau^5}{(1-\overline{\rho}^2)^2}\|B\|\max\{\|S\|, \|R\|\}\frac{Q_{\mathrm{max}}^3}{\mu^3}\right)\Delta \\
&\le \frac{14}{\mu}\frac{\tau^5}{(1-\overline{\rho}^2)^2}\|B\|_+\max\{\|S\|, \|R\|\}\frac{Q_{\mathrm{max}}^3}{\mu^3}\Delta \,.
\end{aligned}
$$

Now suppose that $\Delta$ satisfies:

$$
\begin{aligned}
\Delta &\le \frac{1}{2}\varepsilon\mu(1-\gamma)\left(\frac{\mu}{14}\frac{(1-\overline{\rho}^2)^2}{\tau^5}\frac{1}{\|B\|_+\max\{\|S\|, \|R\|\}}\frac{\mu^3}{Q_{\mathrm{max}}^3}\right) \\
&= \frac{\varepsilon}{28}\mu^2(1-\gamma)\frac{(1-\overline{\rho}^2)^2}{\tau^5}\frac{1}{\|B\|_+\max\{\|S\|, \|R\|\}}\frac{\mu^3}{Q_{\mathrm{max}}^3} \,,
\end{aligned}
\tag{B.8}
$$

we have for all $t \le N_1 - 1$ from (B.5):

$$\delta_\infty(V_{t+1}, V_\star) \le (1-\gamma)\varepsilon/2 + \gamma \cdot \delta_\infty(V_t, V_\star) \,.$$

Unrolling this recursion, we have that for any $N \le N_1$:

$$\delta_\infty(V_N, V_\star) \le \gamma^N \cdot \delta_\infty(V_0, V_\star) + \varepsilon/2 \,.$$

Now observe that for any $N \ge N_0 := \lceil \frac{1}{1-\gamma}\log(2\delta_\infty(V_0, V_\star)/\varepsilon) \rceil$, we obtain:

$$\delta_\infty(V_N, V_\star) \le \varepsilon \,.$$

The claim now follows by combining our four requirements on $\Delta$ given in (B.6), (B.4), (B.7), and (B.8). $\qquad\square$

We now proceed to make several simplifications to Proposition B.2 in order to make the result more presentable. These simplifications come with the tradeoff of introducing extra conservatism into the bounds.

Our first simplification of Proposition B.2 is the following corollary.

**Corollary B.3.** *Consider the sequence of updates defined by Algorithm 4. Suppose we start with a stabilizing $K_0$ and let $V_0$ denote its value function. Define the following variables:*

$$
\begin{aligned}
\mu &:= \min\{\lambda_{\mathrm{min}}(S), \lambda_{\mathrm{min}}(R)\} \,, \\
L &:= \max\{\|S\|, \|R\|\} + 2(\|A\|^2 + \|B\|^2 + 1)\|V_0\|_+ \,, \\
N_0 &:= \lceil (1 + L/\mu)\log(2\delta_\infty(V_0, V_\star)/\varepsilon) \rceil \,.
\end{aligned}
$$

*Fix an $N_1 \ge N_0$ and suppose that*

$$\varepsilon \le \frac{1}{\mu}\left(1 + \frac{L}{\mu}\right)\frac{\|V_0\|}{N_1} \,. \tag{B.9}$$

Suppose the estimates $\widehat{Q}_t$ output by $\mathsf{EstimateQ}$ satisfy, for all $0 \leq t \leq N_1 - 1$, $\widehat{Q}_t \succeq \mu I$ and furthermore,

$$\|\widehat{Q}_t - Q_t\| \leq \frac{\varepsilon}{448} \frac{\mu}{\mu + L} \left(\frac{\mu}{L}\right)^{19/2} .$$

*Then we have for any $N_0 \leq N \leq N_1$ that $\delta_\infty(V_N, V_\star) \leq \varepsilon$. We also have that for any $0 \leq t \leq N_1$, that $A + BK_t$ is $(\sqrt{L/\mu}, \mathsf{Avg}(\sqrt{1 - \mu/L}, 1))$-stable and $\|K_t\| \leq 2L/\mu$.*

*Proof.* First, observe that the map $x \mapsto \frac{x}{\mu + x}$ is increasing, and therefore $\gamma \leq \frac{L}{\mu + L}$ which implies that $1 - \gamma \geq \frac{\mu}{\mu + L}$. Therefore if $\varepsilon \leq \frac{1}{\mu}\left(1 + \frac{L}{\mu}\right)\frac{\|V_0\|}{N_1}$ holds, then we can bound:

$$\min\left\{\frac{\|V_0\|}{N_1}, \varepsilon\mu(1 - \gamma)\right\} \geq \varepsilon\mu\left(\frac{\mu}{\mu + L}\right) .$$

Next, observe that

$$1 - \bar{\rho}^2 = (1 + \bar{\rho})(1 - \bar{\rho}) = (1 + 1/2 + \rho/2)(1/2 - \rho/2) \geq (1 + \rho)(1 - \rho)/4 = (1 - \rho^2)/4 .$$

Therefore,

$$(1 - \bar{\rho}^2)^2 \geq (1 - (1 - \mu/L))^2/16 = (1/16)(\mu/L)^2 .$$

We also have that $\tau \leq \sqrt{\frac{L}{\mu}}$. This means we can bound:

$$\frac{\mu}{28} \frac{(1 - \bar{\rho}^2)^2}{\tau^5} \frac{1}{\|B\|_+ \max\{\|S\|, \|R\|\}} \frac{\mu^3}{Q_{\max}^3} \geq \frac{\mu}{28 \cdot 16}(\mu/L)^{5/2+2}\frac{\mu^3}{L^5} = \frac{1}{448L}\left(\frac{\mu}{L}\right)^{17/2} .$$

Therefore,

$$\min\left\{\frac{\|V_0\|}{N_1}, \varepsilon\mu(1 - \gamma)\right\} \frac{\mu}{28} \frac{(1 - \bar{\rho}^2)^2}{\tau^5} \frac{1}{\|B\|_+ \max\{\|S\|, \|R\|\}} \frac{\mu^3}{Q_{\max}^3} \geq \frac{\varepsilon}{448}\left(\frac{\mu}{\mu + L}\right)\left(\frac{\mu}{L}\right)^{19/2} .$$

The claim now follows from Proposition B.2. $\qquad\square$

Corollary B.3 gives a guarantee in terms of $\delta_\infty(V_N, V_\star) \leq \varepsilon$. By Proposition D.5, this implies a bound on the error of the value functions $\|V_N - V_\star\| \leq \mathcal{O}(\varepsilon)$ for $\varepsilon \leq 1$. In the next corollary, we show we can also control the error $\|K_N - K_\star\| \leq \mathcal{O}(\varepsilon)$.

**Corollary B.4.** *Consider the sequence of updates defined by Algorithm 4. Suppose we start with a stabilizing $K_0$ and let $V_0$ denote its value function. Define the following variables:*

$$\mu := \min\{\lambda_{\min}(S), \lambda_{\min}(R)\} ,$$
$$L := \max\{\|S\|, \|R\|\} + 2(\|A\|^2 + \|B\|^2 + 1)\|V_0\|_+ ,$$
$$N_0 := \left\lceil (1 + L/\mu)\log\left(\frac{2\log(\|V_0\|/\lambda_{\min}(V_\star))}{\varepsilon}\right)\right\rceil .$$

*Suppose that $\varepsilon > 0$ satisfies:*

$$\varepsilon \leq \min\left\{1, \frac{2\log(\|V_0\|/\lambda_{\min}(V_\star))}{e}, \frac{\|V_\star\|^2}{8\mu^2 \log(\|V_0\|/\lambda_{\min}(V_\star))}\right\} .$$

*Suppose we run Algorithm 4 for $N := N_0 + 1$ iterations. Suppose the estimates $\widehat{Q}_t$ output by $\mathsf{EstimateQ}$ satisfy, for all $0 \leq t \leq N_0$, $\widehat{Q}_t \succeq \mu I$ and furthermore,*

$$\|\widehat{Q}_t - Q_t\| \leq \frac{\varepsilon}{448} \frac{\mu}{\mu + L}\left(\frac{\mu}{L}\right)^{19/2} . \tag{B.10}$$

*We have that:*

$$\|K_N - K_\star\| \leq 5\left(\frac{L}{\mu}\right)^2 \varepsilon$$

*and that $A + BK_t$ is $(\sqrt{L/\mu}, \mathsf{Avg}(\sqrt{1 - \mu/L}, 1))$-stable and $\|K_t\| \leq 2L/\mu$ for all $0 \leq t \leq N$.*

*Proof.* We set $N_1 = N_0 + 1$. From this, we compute:

$$
\begin{aligned}
\|K_{N_1} - K_\star\| &= \|G(\widehat{Q}_{N_0}) - G(Q_\star)\| \\
&\stackrel{(a)}{\leq} \frac{(1 + \|G(Q_\star)\|)}{\mu} \|\widehat{Q}_{N_0} - Q_\star\| \\
&\leq \frac{(1 + \|G(Q_\star)\|)}{\mu} (\|\widehat{Q}_{N_0} - Q_{N_0}\| + \|Q_{N_0} - Q_\star\|) \\
&= \frac{(1 + \|G(Q_\star)\|)}{\mu} \left( \|\widehat{Q}_{N_0} - Q_{N_0}\| + \left\| \begin{bmatrix} A^\mathsf{T} \\ B^\mathsf{T} \end{bmatrix} (V_{N_0} - V_\star) \begin{bmatrix} A & B \end{bmatrix} \right\| \right) \\
&\leq \frac{(1 + \|G(Q_\star)\|)}{\mu} (\|\widehat{Q}_{N_0} - Q_{N_0}\| + \|[A \quad B]\|^2 \|V_{N_0} - V_\star\|) \\
&\stackrel{(b)}{\leq} \frac{(1 + \|G(Q_\star)\|)}{\mu} \left( \frac{\varepsilon}{448} \frac{\mu}{\mu + L} \left(\frac{\mu}{L}\right)^{19/2} + \|[A \quad B]\|^2 \|V_{N_0} - V_\star\| \right) \\
&\stackrel{(c)}{\leq} \frac{(1 + \|G(Q_\star)\|)}{\mu} \left( \frac{\varepsilon}{448} \frac{\mu}{\mu + L} \left(\frac{\mu}{L}\right)^{19/2} + e(\|A\|^2 + \|B\|^2)\|V_\star\| \varepsilon \right) \\
&\leq \frac{2L}{\mu^2} \left( \frac{1}{448} \frac{\mu}{\mu + L} \left(\frac{\mu}{L}\right)^{19/2} + 2L \right) \varepsilon \\
&= \left( \frac{1}{224} \frac{1}{\mu + L} \left(\frac{\mu}{L}\right)^{17/2} + 4 \left(\frac{L}{\mu}\right)^2 \right) \varepsilon \\
&\leq 5 \left(\frac{L}{\mu}\right)^2 \varepsilon .
\end{aligned}
$$

Above, (a) follows from Proposition E.3, (b) follows from the bound on $\|\widehat{Q}_{N_0} - Q_{N_0}\|$ from Corollary B.3, and (c) follows from Proposition D.5 and the fact that $\delta_\infty(V_{N_0}, V_\star) \leq \varepsilon$ from Corollary B.3.

Next, we observe that since $V_0 \succeq V_\star$:

$$
\delta_\infty(V_0, V_\star) = \log(\|V_\star^{-1/2} V_0 V_\star^{-1/2}\|) \leq \log(\|V_0\|/\lambda_{\min}(V_\star)) .
$$

Hence we can upper bound $N_0$ from Corollary B.3 by:

$$
N_0 = 2(1 + L/\mu) \log(2 \log(\|P_0\|/\lambda_{\min}(V_\star))/\varepsilon) .
$$

From (B.9), the requirement on $\varepsilon$ is that:

$$
\varepsilon \leq \min \left\{ \frac{\|V_0\|}{2\mu} \frac{1}{\log\left( \frac{2\log(\|V_0\|/\lambda_{\min}(V_\star))}{\varepsilon} \right)}, 1 \right\} .
$$

We will show with Proposition F.3 that a sufficient condition is that:

$$
\varepsilon \leq \min \left\{ 1, \frac{2\log(\|V_0\|/\lambda_{\min}(V_\star))}{e}, \frac{\|V_\star\|^2}{8\mu^2 \log(\|V_0\|/\lambda_{\min}(V_\star))} \right\} .
$$

$\square$

With Corollary B.4 in place, we are now ready to prove Theorem 2.2.

*Proof of Theorem 2.2.* Let $L_0 := A + BK_0$ and let $(\tau, \rho)$ be such that $L_0$ is $(\tau, \rho)$-stable. We know we can pick $\tau = \sqrt{L/\mu}$ and $\rho = \sqrt{1 - \mu/L}$. The covariance $\Sigma_t$ of $x_t$ satisfies:

$$
\Sigma_t = L_0^t \Sigma_0 (L_0^t)^\mathsf{T} + P_t \preceq \tau^2 \rho^{2t} \|\Sigma_0\| I + P_\infty .
$$

Hence for either $t = 0$ or $t \geq \log(\tau)/(1 - \rho)$, $\|\Sigma_t\| \leq \|\Sigma_0\| + \|P_\infty\|$. Therefore, if the trajectory length $T \geq \log(\tau)/(1 - \rho)$, then the operator norm of the initial covariance for every invocation of

LSTD-Q can be bounded by $\|\Sigma_0\| + \|P_\infty\|$, and therefore the proxy variance (2.7) can be bounded by:

$$\bar{\sigma}^2 \leq \tau^2 \rho^4 \|\Sigma_0\| + (1 + \tau^2 \rho^4)\|P_\infty\| + \sigma_\eta^2 \|B\|^2$$
$$\leq 2(L/\mu)(\|\Sigma_0\| + \|P_\infty\| + \sigma_\eta^2\|B\|^2) \,.$$

By Corollary B.4, when condition (B.10) holds, we have that $A + BK_t$ is $(\tau, \mathsf{Avg}(\rho, 1))$ stable, $\|K_t\| \leq 2L/\mu$, and $\|Q_t\| \leq L$ for all $0 \leq t \leq N_0 + 1$. We now define $\bar{\varepsilon} := 5(L/\mu)^2 \varepsilon$. If we can ensure that

$$\|\widehat{Q}_t - Q_t\| \leq \frac{1}{2240}\left(\frac{\mu}{\mu + L}\right)\left(\frac{\mu}{L}\right)^{23/2} \bar{\varepsilon} \,, \tag{B.11}$$

then if

$$\bar{\varepsilon} \leq 5\left(\frac{L}{\mu}\right)^2 \min\left\{1, \frac{2\log(\|V_0\|/\lambda_{\min}(V_\star))}{e}, \frac{\|V_\star\|^2}{8\mu^2\log(\|V_0\|/\lambda_{\min}(V_\star))}\right\} \,,$$

then by Corollary B.4 we ensure that $\|K_N - K\| \leq \bar{\varepsilon}$. By Theorem 2.1, (B.11) can be ensured by first observing that $Q_t \succeq \mu I$ and therefore for any symmetric $\widehat{Q}$ we have:

$$\|\mathsf{Proj}_\mu(\widehat{Q}) - Q_t\| \leq \|\mathsf{Proj}_\mu(\widehat{Q}) - Q_t\|_F \leq \|\widehat{Q} - Q_t\|_F \,.$$

Above, the last inequality holds because $\mathsf{Proj}_\mu(\cdot)$ is the Euclidean projection operator associated with $\|\cdot\|_F$ onto the convex set $\{Q : Q \succeq \mu I, \ Q = Q^\mathsf{T}\}$. Now combining (2.9) and (2.8) and using the bound $\frac{\tau^2}{\rho^2(1-\rho^2)} \leq \frac{(L/\mu)^2}{1-\mu/L}$:

$$T \geq \widetilde{O}(1) \max\left\{(n+d)^2, \right.$$
$$\frac{L^2}{(1-\mu/L)^2}\left(\frac{L}{\mu}\right)^{17}\frac{(n+d)^4}{\sigma_\eta^4}\sigma_w^2(\|\Sigma_0\| + \|P_\infty\| + \sigma_\eta^2\|B\|^2),$$
$$\left.\frac{1}{\bar{\varepsilon}^2}\frac{L^4}{(1-\mu/L)^2}\left(\frac{L}{\mu}\right)^{42}\frac{(n+d)^3}{\sigma_\eta^4}\sigma_w^2(\|\Sigma_0\| + \|P_\infty\| + \sigma_\eta^2\|B\|^2)\right\} \,.$$

Theorem 2.2 now follows. $\qquad\square$

## C   Analysis for Adaptive LSPI

In this section we develop our analysis for Algorithm 3. We start by presenting a meta adaptive algorithm (Algorithm 5) and lay out sufficient conditions for the meta algorithm to achieve sub-linear regret. We then specialize the meta algorithm to use LSPI as a sub-routine.

---

**Algorithm 5** General Adaptive LQR Algorithm

---

**Input:** Initial stabilizing controller $K^{(0)}$, number of epochs $E$, epoch multiplier $T_{\mathrm{mult}}$.
1: **for** $i = 0, ..., E - 1$ **do**
2:     Set $T_i = T_{\mathrm{mult}}2^i$.
3:     Set $\sigma_{\eta,i}^2 = \sigma_w^2\left(\frac{1}{2^i}\right)^{1/(1+\alpha)}$.
4:     Roll system forward $T_i$ steps with input $u_t^{(i)} = K^{(i)}x_t^{(i)} + \eta_t^{(i)}$, where $\eta_t^{(i)} \sim \mathcal{N}(0, \sigma_{\eta,i}^2 I)$.
5:     Let $\mathcal{D}_i = \{(x_t^{(i)}, u_t^{(i)}, x_{t+1}^{(i)})\}_{t=0}^{T_i}$.
6:     Set $K^{(i+1)} = \mathsf{EstimateK}(K^{(i)}, \mathcal{D}_i)$.
7: **end for**

---

Algorithm 5 is the general form of the $\varepsilon$-greedy strategy for adaptive LQR recently described in Dean et al. [13] and Mania et al. [26]. We study Algorithm 5 under the following assumption regarding the sub-routine EstimateK.

**Assumption 1.** *We assume there exists two functions* $C_{\mathrm{req}}, C_{\mathrm{err}}$ *and* $\alpha \geq 1$ *such that the following holds. Suppose the controller* $K^{(i)}$ *that generates* $\mathcal{D}_i$ *stabilizes* $(A, B)$ *and* $V^{(i)}$ *is its associated value function, the initial condition* $x_0^{(i)} \sim \mathcal{N}(0, \Sigma_0^{(i)})$, *and that the trajectory* $\mathcal{D}_i$ *is collected via* $u_t^{(i)} = K^{(i)} x_t^{(i)} + \eta_t^{(i)}$ *with* $\eta_t^{(i)} \sim \mathcal{N}(0, \sigma_{\eta,i}^2 I)$. *For any* $0 < \varepsilon < C_{\mathrm{req}}(\|V^{(i)}\|)$ *and any* $\delta \in (0, 1)$, *as long as* $|\mathcal{D}_i|$ *satisfies:*

$$|\mathcal{D}_i| \geq \frac{C_{\mathrm{err}}(\|V^{(i)}\|, \|\Sigma_0^{(i)}\|)}{\varepsilon^2} \frac{1}{\sigma_{\eta,i}^{2\alpha}} \operatorname{polylog}(|\mathcal{D}_i|, 1/\sigma_{\eta,i}^\alpha, 1/\delta, 1/\varepsilon) , \qquad \text{(C.1)}$$

*then we have with probability at least* $1 - \delta$ *that* $\|K^{(i+1)} - K_\star\| \leq \varepsilon$. *We also assume the function* $C_{\mathrm{req}}$ *(resp.* $C_{\mathrm{err}}$*) is monotonically decreasing (resp. increasing) with respect to its arguments, and that the functions are allowed to depend in any way on the problem parameters* $(A, B, S, R, n, d, \sigma_w^2, K_\star, P_\star)$

Before turning to the analysis of Algorithm 5, we state a simple proposition that bounds the covariance matrix along the trajectory induced by Algorithm 5.

**Proposition C.1.** *Fix a* $j \geq 1$. *Let* $\Sigma_0^{(j)}$ *denote the covariance matrix of* $x_0^{(j)}$. *Suppose that for all* $0 \leq i < j$ *each* $K^{(i)}$ *stabilizes* $(A, B)$ $A + BK^{(i)}$ *is* $(\tau, \rho)$*-stable. Also suppose that* $\sigma_{\eta,i} \leq \sigma_w$ *and that*

$$T_{\mathrm{mult}} \geq \frac{1}{2(1-\rho)} \log\left(\frac{n\tau^2}{\rho^2}\right) .$$

*We have that:*

$$\operatorname{tr}(\Sigma_0^{(j)}) \leq \sigma_w^2 (1 + \|B\|^2) n \frac{\tau^2}{(1 - \rho^2)^2} .$$

*Proof.* Let $L_i = A + BK^{(i)}$. We write:

$$
\begin{aligned}
\mathbb{E}[\|x_0^{(i)}\|^2] &= \mathbb{E}[\mathbb{E}[\|x_0^{(i)}\|^2 | x_0^{(i-1)}]] \\
&= \mathbb{E}[\mathbb{E}[\operatorname{tr}(x_0^i (x_0^i)^\mathsf{T}) | x_0^{(i-1)}]] \\
&\leq \mathbb{E}[\operatorname{tr}(L_{i-1}^{T_{i-1}} x_0^{(i-1)} (x_0^{(i-1)})^\mathsf{T} (L_{i-1}^{T_{i-1}})^\mathsf{T})] + (\sigma_w^2 + \sigma_{\eta,i-1}^2 \|B\|^2) n \frac{\tau^2}{1 - \rho^2} \\
&\leq n\tau^2 \rho^{2T_{i-1}} \mathbb{E}[\|x_0^{(i-1)}\|^2] + \sigma_w^2 (1 + \|B\|^2) n \frac{\tau^2}{1 - \rho^2} .
\end{aligned}
$$

We have that $x_0^{(0)} = 0$. Hence if we choose $T_{\mathrm{mult}}$ such that $n\tau^2 \rho^{2T_{\mathrm{mult}}} \leq \rho^2$, we obtain the recurrence:

$$\mathbb{E}[\|x_0^{(i)}\|^2] \leq \rho^2 \mathbb{E}[\|x_0^{(i-1)}\|^2] + \sigma_w^2 (1 + \|B\|^2) n \frac{\tau^2}{1 - \rho^2} ,$$

and therefore $\mathbb{E}[\|x_0^{(i)}\|^2] \leq \sigma_w^2 (1 + \|B\|^2) n \frac{\tau^2}{(1-\rho^2)^2}$ for all $i$. This is ensured if

$$T_{\mathrm{mult}} \geq \frac{1}{2(1-\rho)} \log(n\tau^2/\rho^2) .$$

$\square$

Next, we state a lemma that relates the instantaneous cost to the expected cost. The proof is based on the Hanson-Wright inequality, and appears in Dean et al. [13]. Let the notation $J(K; \Sigma)$ denote the infinite horizon average LQR cost when the feedback $u_t = Kx_t$ is played and when the process noise is $w_t \sim \mathcal{N}(0, \Sigma)$. Explicitly:

$$J(K; \Sigma) = \operatorname{tr}(\Sigma V(K)) , \quad V(K) = \mathsf{dlyap}(A + BK, S + K^\mathsf{T} RK) . \qquad \text{(C.2)}$$

With this notation, we have the following lemma.

**Lemma C.2** (Lemma D.2, [13]). *Let $x_0 \sim \mathcal{N}(0, \Sigma_0)$ and suppose that $u_t = Kx_t + \eta_t$ with $A + BK$ as $(\tau, \rho)$-stable and $\eta_t \sim \mathcal{N}(0, \sigma_\eta^2 I)$. We have that with probability at least $1 - \delta$:*

$$\sum_{t=1}^{T} x_t^\mathsf{T} Q x_t + u_t^\mathsf{T} R u_t \le T J(K; \sigma_w^2 I + \sigma_\eta^2 BB^\mathsf{T})$$

$$+ c\sqrt{nT} \frac{\tau^2}{(1-\rho)^2} (\|\Sigma_0\| + \sigma_w^2 + \sigma_\eta^2 \|B\|^2) \|Q + K^\mathsf{T} R K\| \log(1/\delta) .$$

Finally, we state a second order perturbation result from Fazel et al. [16], which was recently used by Mania et al. [26] to study certainty equivalent controllers.

**Lemma C.3** (Lemma 12, [16]). *Let $K$ stabilize $(A, B)$ with $A + BK$ as $(\tau, \rho)$-stable, and let $K_\star$ be the optimal LQR controller for $(A, B, Q, R)$ and $V_\star$ be the optimal value function. We have that:*

$$J(K) - J_\star \le \sigma_w^2 \frac{\tau^2}{1 - \rho^2} \|R + B^\mathsf{T} V_\star B\| \|K - K_\star\|_F^2 .$$

With these tools in place, we are ready to state our main result regarding the regret incurred (c.f. (2.16)) by Algorithm 5.

**Proposition C.4.** *Fix a $\delta \in (0, 1)$. Suppose that* EstimateK *satisfies Assumption 1. Let the initial feedback $K^{(0)}$ stabilize $(A, B)$ and let $V^{(0)}$ denote its associated value function. Also let $K_\star$ denote the optimal LQR controller and let $V_\star$ denote the optimal value function. Let $\Gamma_\star = 1 + \max\{\|A\|, \|B\|, \|V^{(0)}\|, \|V_\star\|, \|K^{(0)}\|, \|K_\star\|, \|Q\|, \|R\|\}$. Define the following bounds:*

$$K_{\max} := \Gamma_\star ,$$

$$V_{\max} := 4 \frac{\Gamma_\star^5}{\lambda_{\min}(S)^2} ,$$

$$\Sigma_{\max} := 4\sigma_w^2 n \frac{\Gamma_\star^4}{\lambda_{\min}(S)^2} .$$

*Suppose that $T_{\mathrm{mult}}$ satisfies:*

$$T_{\mathrm{mult}} \ge \max\left\{1, \frac{\Gamma_\star^8}{\lambda_{\min}(S)^4}, \frac{1}{C_{\mathrm{req}}^4(V_{\max})}\right\} \frac{C_{\mathrm{err}}^2(V_{\max}, \Sigma_{\max})}{\sigma_w^4} \mathrm{poly}(\alpha) \mathrm{polylog}(1/\sigma_w, E/\delta) .$$

*With probability at least $1 - \delta$, we have that:*

$$\mathsf{Regret}(T) \le \sigma_w^{2(1-\alpha)} d \frac{\Gamma_\star^7}{\lambda_{\min}(S)^2} C_{\mathrm{err}}^2(V_{\max}, \Sigma_{\max}) \left(\frac{T+1}{T_{\mathrm{mult}}}\right)^{\alpha/(\alpha+1)} \mathrm{polylog}(T/\delta)$$

$$+ T_{\mathrm{mult}} \Gamma_\star^2 J_\star \left(\frac{T+1}{T_{\mathrm{mult}}}\right)^{\alpha/(\alpha+1)}$$

$$+ \mathcal{O}(1) n^{3/2} \sqrt{T} \sigma_w^2 \frac{\Gamma_\star^9}{\lambda_{\min}(S)^4} \log(T/\delta) + o_T(1) .$$

*Proof.* We state the proof assuming that $T$ is at an epoch boundary for simplicity. Each epoch has length $T_i = T_{\mathrm{mult}} 2^i$. Let $T_0 + T_1 + ... + T_{E-1} = T$. This means that $E = \log_2((T+1)/T_{\mathrm{mult}})$.

We start by observing that by Proposition E.4, we have that $A + BK^{(0)}$ is $(\tau, \rho)$-stable for $\tau := \sqrt{\|V^{(0)}\|/\lambda_{\min}(S)}$ and $\rho := \sqrt{1 - \lambda_{\min}(S)/\|V^{(0)}\|}$. We will show that $A + BK^{(i)}$ is $(\tau, \overline{\rho})$-stable for $i = 1, ..., E - 1$ for $\overline{\rho} := \mathsf{Avg}(\rho, 1)$. By Lemma E.1, this occurs if we can ensure that $\|K^{(i)} - K_\star\| \le \frac{(1-\rho)}{2\tau\|B\|}$. for $i = 1, ..., E - 1$.

We will also construct bounds $K_{\max}, V_{\max}, \Sigma_{\max}$ such that $\|K^{(i)}\| \le K_{\max}$, $\|V^{(i)}\| \le V_{\max}$, and $\|\Sigma^{(i)}\| \le \Sigma_{\max}$ for all $0 \le i \le E - 1$. We set the bounds as:

$$K_{\max} := \max\{\|K^{(0)}\|, \|K_\star\| + 1\} ,$$

$$V_{\max} := \max\{\|V^{(0)}\|, \frac{\tau^2}{1 - \overline{\rho}^2} (\|Q\| + \|R\| K_{\max}^2)\} ,$$

$$\Sigma_{\max} := \sigma_w^2 (1 + \|B\|^2) n \frac{\tau^2}{1 - \overline{\rho}^2} .$$

In what follows, we will use the shorthand $C_{\mathrm{req}} = C_{\mathrm{req}}(V_{\max})$ and $C_{\mathrm{err}} = C_{\mathrm{err}}(V_{\max}, \Sigma_{\max})$.

Before we continue, we first argue that our choice of $T_{\mathrm{mult}}$ satisfies for all $i = 1, ..., E - 1$:

$$T_{i-1} \geq \max\{1, \frac{\tau^2\|B\|^2}{4(1-\overline{\rho})^2}, \frac{1}{C_{\mathrm{req}}^2}\} \frac{C_{\mathrm{err}}}{\sigma_{\eta,i}^{2\alpha}} \operatorname{polylog}(T_{i-1}, 1/\sigma_{i,\eta}^{\alpha}, 1/\sigma_w, E/\delta) . \tag{C.3}$$

Rearranging, this is equivalent to:

$$T_{\mathrm{mult}} \geq \max\{1, \frac{\tau^2\|B\|^2}{4(1-\overline{\rho})^2}, \frac{1}{C_{\mathrm{req}}^2}\} 2C_{\mathrm{err}}\sigma_w^{-2} \frac{1}{(2^i)^{1/(1+\alpha)}} \operatorname{polylog}(T_{\mathrm{mult}}2^i, (2^i)^{\alpha/(1+\alpha)}, 1/\sigma_w, E/\delta) .$$

We first remove the dependence on $i$ on the RHS by taking the maximum over all $i$. By Proposition F.2, it suffices to take $T_{\mathrm{mult}}$ satisfying:

$$T_{\mathrm{mult}} \geq \max\{1, \frac{\tau^2\|B\|^2}{4(1-\overline{\rho})^2}, \frac{1}{C_{\mathrm{req}}^2}\} \frac{C_{\mathrm{err}}}{\sigma_w^2} \operatorname{poly}(\alpha) \operatorname{polylog}(T_{\mathrm{mult}}, 1/\sigma_w, E/\delta) .$$

We now remove the implicit dependence on $T_{\mathrm{mult}}$. By Proposition F.4, it suffices to take $T_{\mathrm{mult}}$ satisfying:

$$\begin{aligned} T_{\mathrm{mult}} \geq \max\{1, \frac{\tau^2\|B\|^2}{4(1-\overline{\rho})^2}, \frac{1}{C_{\mathrm{req}}^2}\} \frac{C_{\mathrm{err}}}{\sigma_w^2} \\ \times \operatorname{poly}(\alpha) \operatorname{polylog}(1/\sigma_w, E/\delta, \tau, \|B\|, 1/(1-\overline{\rho}), 1/C_{\mathrm{req}}, C_{\mathrm{err}}) . \end{aligned}$$

We are now ready to proceed.

First we look at the base case $i = 0$. Clearly, the bounds work for $i = 0$ by definition. Now we look at epoch $i \geq 1$ and we assume the bounds hold for $\ell = 0, ..., i - 1$. For $i \geq 1$ we define $\varepsilon_i$ as:

$$\varepsilon_i := \inf\left\{\varepsilon \in (0,1) : T_{i-1} \geq \frac{C_{\mathrm{err}}}{\varepsilon^2} \frac{1}{\sigma_{\eta,i}^{2\alpha}} \operatorname{polylog}(T_{i-1}, 1/\sigma_{\eta,i}^{\alpha}, E/\delta, 1/\varepsilon)\right\} .$$

By Proposition F.1, we have that as long as

$$T_{i-1} \geq C_{\mathrm{err}} \frac{1}{\sigma_{\eta,i}^{2\alpha}} \operatorname{polylog}(T_{i-1}, 1/\sigma_{\eta,i}^{\alpha}, E/\delta) , \tag{C.4}$$

then we have that $\varepsilon_i$ satisfies:

$$\varepsilon_i^2 \leq \frac{C_{\mathrm{err}}}{T_{i-1}\sigma_{\eta,i}^{2\alpha}} \operatorname{polylog}(T_{i-1}, 1/\sigma_{\eta,i}^{\alpha}, E/\delta) . \tag{C.5}$$

But (C.4) is implied by (C.3), so we know that (C.5) holds. Therefore, we have $\|K^{(i)} - K_\star\| \leq \varepsilon_i$. Now by (C.5), if:

$$\frac{C_{\mathrm{err}}}{T_{i-1}\sigma_{\eta,i}^{2\alpha}} \operatorname{polylog}(T_{i-1}, 1/\sigma_{\eta,i}^{\alpha}, E/\delta) \leq \min\{1, \frac{(1-\overline{\rho})^2}{4\tau^2\|B\|^2}, C_{\mathrm{req}}^2\} , \tag{C.6}$$

then the following is true:

$$\varepsilon_i \leq \min\{1, (1-\overline{\rho})/(2\tau\|B\|), C_{\mathrm{req}}\} .$$

However, (C.6) is also implied by (C.3), so we have by Assumption 1:

$$\|K^{(i)} - K_\star\| \leq \min\{1, (1-\overline{\rho})/(2\tau\|B\|)\} .$$

This has several implications. First, it implies that:

$$\|K^{(i)}\| \leq \|K_\star\| + 1 \leq K_{\max} .$$

Next, it implies by Lemma E.1 that $A + BK^{(i)}$ is $(\tau, \overline{\rho})$-stable. Next, by Proposition C.1, it implies that $\|\Sigma^{(i)}\| \leq \Sigma_{\max}$. Finally, letting $L_i := A + BK^{(i)}$, we have that:

$$\begin{aligned} \|V^{(i)}\| &= \left\|\sum_{\ell=0}^{\infty} (L_i)^\ell (Q + (K^{(i)})^{\mathsf{T}} R K^{(i)})(L_i^{\mathsf{T}})^\ell\right\| \\ &\leq \frac{\tau^2}{1-\overline{\rho}^2}(\|Q\| + \|R\|K_{\max}^2) \\ &\leq V_{\max} . \end{aligned}$$

Thus, by induction we have that $\|K^{(i)}\| \leq K_{\max}$, $\|V^{(i)}\| \leq V_{\max}$, and $\|\Sigma^{(i)}\| \leq \Sigma_{\max}$ for all $0 \leq i \leq E - 1$.

We are now ready to bound the regret. From (C.2), we see the relation $J(K; \sigma_w^2 I + \sigma_\eta^2 BB^\mathsf{T}) \leq \left(1 + \frac{\sigma_\eta^2 \|B\|^2}{\sigma_w^2}\right) J(K; \sigma_w^2 I)$ holds. Therefore by Lemma C.2 and Lemma C.3,

$$\sum_{t=1}^{T} x_t^\mathsf{T} Q x_t + u_t^\mathsf{T} R u_t - T J_\star \leq T\left(1 + \frac{\sigma_\eta^2 \|B\|^2}{\sigma_w^2}\right)\left(J_\star + \sigma_w^2 \frac{\tau^2}{1 - \bar{\rho}^2}\|R + B^\mathsf{T} P_\star B\|\|K - K_\star\|_F^2\right) - T J_\star$$

$$+ c\sqrt{nT}\frac{\tau^2}{(1 - \bar{\rho})^2}(\|P_0\| + \sigma_w^2 + \sigma_\eta^2 \|B\|^2)\|Q + K^\mathsf{T} R K\| \log(1/\delta)$$

$$\leq T(\sigma_w^2 + \sigma_\eta^2 \|B\|^2)\frac{\tau^2}{1 - \bar{\rho}^2}(\|R\| + \|P_\star\|\|B\|^2)\|K - K_\star\|_F^2 + T\frac{\sigma_\eta^2 \|B\|^2}{\sigma_w^2} J_\star$$

$$+ c\sqrt{nT}\frac{\tau^2}{(1 - \bar{\rho})^2}(\|P_0\| + \sigma_w^2 + \sigma_\eta^2 \|B\|^2)(\|Q\| + \|K\|^2\|R\|) \log(1/\delta).$$

Using the inequality above,

$$\mathsf{Regret}(T) = \sum_{i=0}^{E-1} \sum_{t=1}^{T_i} (x_t^{(i)})^\mathsf{T} Q(x_t^{(i)}) + (u_t^{(i)})^\mathsf{T} R(u_t^{(i)}) - T J_\star$$

$$\leq \sum_{i=0}^{E-1} T_i \sigma_w^2(1 + \|B\|^2)\frac{\tau^2}{1 - \bar{\rho}^2}(\|R\| + \|V_\star\|\|B\|^2)\|K^{(i)} - K_\star\|_F^2 + T_i \frac{\sigma_{\eta,i}^2 \|B\|^2}{\sigma_w^2} J_\star$$

$$+ c\sqrt{nT_i}\sigma_w^2(1 + \|B\|^2)n\frac{\tau^4}{(1 - \bar{\rho}^2)^4}(\|Q\| + K_{\max}^2 \|R\|) \log(E/\delta)$$

$$\leq \mathcal{O}(1) + \sum_{i=1}^{E-1} \sigma_w^2(1 + \|B\|^2)\frac{d\tau^2}{1 - \bar{\rho}^2}(\|R\| + \|V_\star\|\|B\|^2)C_{\mathrm{err}}^2 \frac{2}{\sigma_{\eta,i}^{2\alpha}} \mathrm{polylog}(E/\delta, 1/\sigma_{\eta,i})$$

$$+ T_i \frac{\sigma_{\eta,i}^2 \|B\|^2}{\sigma_w^2} J_\star$$

$$+ c\sqrt{nT_i}\sigma_w^2(1 + \|B\|^2)n\frac{\tau^4}{(1 - \bar{\rho}^2)^4}(\|Q\| + K_{\max}^2 \|R\|) \log(E/\delta)$$

$$= 2\sum_{i=1}^{E-1} \sigma_w^{2-2\alpha}(1 + \|B\|^2)\frac{d\tau^2}{1 - \bar{\rho}^2}(\|R\| + \|V_\star\|\|B\|^2)C_{\mathrm{err}}(2^i)^{\alpha/(1+\alpha)} \mathrm{polylog}(E/\delta, 1/\sigma_{\eta,i})$$

$$+ T_{\mathrm{mult}}(2^i)^{\alpha/(1+\alpha)}\|B\|^2 J_\star$$

$$+ c\sqrt{nT_i}\sigma_w^2(1 + \|B\|^2)n\frac{\tau^4}{(1 - \bar{\rho}^2)^4}(\|Q\| + K_{\max}^2 \|R\|) \log(E/\delta) + \mathcal{O}(1)$$

$$\leq \sigma_w^{2-2\alpha}(1 + \|B\|^2)\frac{d\tau^2}{1 - \bar{\rho}^2}(\|R\| + \|V_\star\|\|B\|^2)C_{\mathrm{err}}^2 \frac{\alpha + 1}{\alpha}\left(\frac{T + 1}{T_{\mathrm{mult}}}\right)^{\alpha/(\alpha+1)} \mathrm{polylog}(T/\delta)$$

$$+ T_{\mathrm{mult}}\|B\|^2 J_\star \frac{\alpha + 1}{\alpha}\left(\frac{T + 1}{T_{\mathrm{mult}}}\right)^{\alpha/(\alpha+1)}$$

$$+ \mathcal{O}(1)\sqrt{nT}\sigma_w^2(1 + \|B\|^2)n\frac{\tau^4}{(1 - \bar{\rho}^2)^4}(\|Q\| + K_{\max}^2 \|R\|) \log(T/\delta) + \mathcal{O}(1).$$

The last inequality holds because:

$$\sum_{i=1}^{E-1} (2^i)^{\alpha/(1+\alpha)} \leq \int_1^E (2^x)^{\alpha/(1+\alpha)} \, dx \leq \frac{1}{\log 2}\frac{\alpha + 1}{\alpha}(2^E)^{\alpha/(\alpha+1)} = \frac{1}{\log 2}\frac{\alpha + 1}{\alpha}\left(\frac{T + 1}{T_{\mathrm{mult}}}\right)^{\alpha/(\alpha+1)}.$$

Now observe that we can bound

$$K_{\max} \leq \Gamma_\star \,,$$

$$V_{\max} \leq 4 \frac{\tau^2}{1 - \rho^2} \Gamma_\star^3 \,,$$

$$\Sigma_{\max} \leq 4 \sigma_w^2 n \Gamma_\star^2 \frac{\tau^2}{1 - \rho^2} \,,$$

$$\frac{\tau^2}{1 - \rho^2} \leq \frac{\Gamma_\star^2}{\lambda_{\min}(S)^2} \,.$$

Therefore:

$$
\begin{aligned}
\mathsf{Regret}(T) \leq{}& \sigma_w^{2(1-\alpha)} d \frac{\Gamma_\star^7}{\lambda_{\min}(S)^2} C_{\mathrm{err}}^2 \left( \frac{T+1}{T_{\mathrm{mult}}} \right)^{\alpha/(\alpha+1)} \mathrm{polylog}(T/\delta) \\
&+ T_{\mathrm{mult}} \Gamma_\star^2 J_\star \left( \frac{T+1}{T_{\mathrm{mult}}} \right)^{\alpha/(\alpha+1)} \\
&+ \mathcal{O}(1) n^{3/2} \sqrt{T} \sigma_w^2 \frac{\Gamma_\star^9}{\lambda_{\min}(S)^4} \log(T/\delta) + o_T(1) \,.
\end{aligned}
$$

$\square$

We now turn to the proof of Theorem 2.3 and analyze Algorithm 3 by applying Proposition C.4 with LSPI (Section B) taking the place of EstimateK. To apply Proposition C.4, we use Theorem 2.2 to compute the bounds $C_{\mathrm{req}}, C_{\mathrm{err}}$ that are needed for Assumption 1 to hold. The following proposition will be used to work out these bounds.

**Proposition C.5.** *Let $P_1 = \mathsf{dlyap}(L, M_1)$ and $P_2 = \mathsf{dlyap}(L^\mathsf{T}, M_2)$, and suppose both $M_1$ and $M_2$ are $n \times n$ positive definite. We have that:*

$$\|P_1\| \leq n \frac{\|M_1\|}{\sigma_{\min}(M_2)} \|P_2\| \,.$$

*Proof.* We start with the observation that $\mathrm{tr}(M_2 P_1) = \mathrm{tr}(M_1 P_2)$. Then we lower bound $\mathrm{tr}(M_2 P_1) \geq \sigma_{\min}(M_2) \mathrm{tr}(P_1) \geq \sigma_{\min}(M_2) \|P_1\|$, and upper bound $\mathrm{tr}(M_1 P_2) \leq \|M_1\| \mathrm{tr}(P_2) \leq n \|M_1\| \|P_2\|$. $\square$

We use Proposition C.5 to compute the following upper bound for $P_\infty$:

$$\|P_\infty\| \leq n \frac{\sigma_w^2 + \sigma_\eta^2 \|B\|^2}{\lambda_{\min}(S)} \|V_\star\| \leq \sigma_w^2 n \frac{\Gamma_\star^2}{\lambda_{\min}(S)} \,.$$

We first compute the $C_{\mathrm{req}}$ term from (2.12):

$$C_{\mathrm{req}}(\|V^{(i)}\|) = \min \left\{ 1, \frac{2 \log(\|V^{(i)}\|/\lambda_{\min}(V_\star))}{e}, \frac{\|V_\star\|^2}{8\mu^2 \log(\|V^{(i)}\|/\lambda_{\min}(V_\star))} \right\} \,.$$

We see that $C_{\mathrm{req}}$ is monotonically decreasing in $\|V^{(i)}\|$.

Next we compute $C_{\mathrm{err}}$ from (2.14). First we see that $\alpha = 2$. Observing we can upper bound $L \leq \Gamma_\star^2 \|V^{(i)}\|_+$, we have that:

$$C_{\mathrm{err}}(\|V^{(i)}\|, \|\Sigma_0^{(i)}\|) = \frac{\Gamma_\star^{94}}{(1 - \mu/(\Gamma_\star^2 \|V_\star\|_+))^2} \frac{\|V^{(i)}\|_+^{47}}{\mu^{43}} (n+d)^4 \sigma_w^2 \left( \|\Sigma_0^{(i)}\| + \sigma_w^2 n \frac{\Gamma_\star^2}{\lambda_{\min}(S)} + \sigma_w^2 \Gamma_\star^2 \right) \,.$$

We see that $C_{\mathrm{err}}$ is monotonically increasing in both $\|V^{(i)}\|$ and $\|\Sigma_0^{(i)}\|$. This gives the proof of Theorem 2.3.

# D  Properties of the Invariant Metric

Here we review relevant properties of the invariant metric $\delta_\infty(A, B) = \|\log(A^{-1/2}BA^{-1/2})\|$ over positive definite matrices.

**Lemma D.1** (c.f. [22]). *Suppose that $A$ is positive semidefinite and $X, Y$ are positive definite. Also suppose that $M$ is invertible. We have:*

*(i) $\delta_\infty(X, Y) = \delta_\infty(X^{-1}, Y^{-1}) = \delta_\infty(MXM^\mathsf{T}, MYM^\mathsf{T})$.*

*(ii) $\delta_\infty(A + X, A + Y) \leq \frac{\alpha}{\alpha+\beta}\delta_\infty(X, Y)$, where $\alpha = \max\{\lambda_{\max}(X), \lambda_{\max}(Y)\}$ and $\beta = \lambda_{\min}(A)$.*

**Lemma D.2** (c.f. Theorem 4.4, [22]). *Consider the map $f(X) = A + M(B + X^{-1})^{-1}M^\mathsf{T}$, where $A, B$ are PSD and $X$ is positive definite. Suppose that $X, Y$ are two positive definite matrices and $A$ is invertible. We have:*

$$\delta_\infty(f(X), f(Y)) \leq \frac{\max\{\lambda_1(MXM^\mathsf{T}), \lambda_1(MYM^\mathsf{T})\}}{\lambda_{\min}(A) + \max\{\lambda_1(MXM^\mathsf{T}), \lambda_1(MYM^\mathsf{T})\}}\delta_\infty(X, Y)\,.$$

*Proof.* We first assume that $M$ is invertible. Using the properties of $\delta_\infty$ from Lemma D.1, we have:

$$\begin{aligned}
\delta_\infty(f(X), f(Y)) &= \delta_\infty(A + M(B + X^{-1})^{-1}M^\mathsf{T}, A + M(B + Y^{-1})^{-1}M^\mathsf{T}) \\
&\leq \frac{\alpha}{\lambda_{\min}(A) + \alpha}\delta_\infty(M(B + X^{-1})^{-1}M^\mathsf{T}, M(B + Y^{-1})^{-1}M^\mathsf{T}) \\
&= \frac{\alpha}{\lambda_{\min}(A) + \alpha}\delta_\infty((B + X^{-1})^{-1}, (B + Y^{-1})^{-1}) \\
&= \frac{\alpha}{\lambda_{\min}(A) + \alpha}\delta_\infty(B + X^{-1}, B + Y^{-1}) \\
&\leq \frac{\alpha}{\lambda_{\min}(A) + \alpha}\delta_\infty(X^{-1}, Y^{-1}) \\
&= \frac{\alpha}{\lambda_{\min}(A) + \alpha}\delta_\infty(X, Y)\,,
\end{aligned}$$

where $\alpha = \max\{\lambda_{\max}(M(B + X^{-1})^{-1}M^\mathsf{T}), \lambda_{\max}(M(B + X^{-1})^{-1}M^\mathsf{T})\}$. Now, we observe that:

$$B + X^{-1} \succeq X^{-1} \iff (B + X^{-1})^{-1} \preceq X\,.$$

This means that $M(B + X^{-1})^{-1}M^\mathsf{T} \preceq MXM^\mathsf{T}$ and similarly $M(B + Y^{-1})^{-1}M^\mathsf{T} \preceq MYM^\mathsf{T}$. This proves the claim when $M$ is invertible. When $M$ is not invertible, use a standard limiting argument. $\square$

**Proposition D.3.** *Suppose that $A, B$ are positive definite matrices satisfying $A \succeq \mu I$, $B \succeq \mu I$. We have that:*

$$\delta_\infty(A, B) \leq \frac{\|A - B\|}{\mu}\,.$$

*Proof.* We have that:

$$\|A^{-1/2}BA^{-1/2}\| = \|A^{-1/2}(B - A)A^{-1/2} + I\| \leq 1 + \frac{\|B - A\|}{\mu}\,.$$

Taking log on both sides and using $\log(1 + x) \leq x$ for $x \geq 0$ yields the claim. $\square$

**Proposition D.4.** *Suppose that $B \preceq A_1 \preceq A_2$ are all positive definite matrices. We have that:*

$$\delta_\infty(A_1, B) \leq \delta_\infty(A_2, B)\,.$$

*Proof.* The chain of orderings implies that:

$$I \preceq B^{-1/2}A_1B^{-1/2} \preceq B^{-1/2}A_2B^{-1/2}\,.$$

Therefore:

$$\delta_\infty(A_1, B) = \log \lambda_{\max}(B^{-1/2} A_1 B^{-1/2}) \le \log \lambda_{\max}(B^{-1/2} A_2 B^{-1/2}) = \delta_\infty(A_2, B) \,.$$

Each step requires careful justification. The first equality holds because $I \preceq B^{-1/2} A_1 B^{-1/2}$ and the second inequality uses the monotonicity of the scalar function $x \mapsto \log x$ on $\mathbb{R}_+$ in addition to $B^{-1/2} A_1 B^{-1/2} \preceq B^{-1/2} A_2 B^{-1/2}$. $\qquad\square$

**Proposition D.5.** *Suppose that $A, B$ are positive definite matrices with $B \succeq A$. We have that:*

$$\|A - B\| \le \|A\|(\exp(\delta_\infty(A, B)) - 1) \,.$$

*Furthermore, if $\delta_\infty(A, B) \le 1$ we have:*

$$\|A - B\| \le e\|A\|\delta_\infty(A, B) \,.$$

*Proof.* The assumption that $B \succeq A$ implies that $A^{-1/2} B A^{-1/2} \succeq I$ and that $\|A - B\| = \lambda_{\max}(B - A)$. Now observe that:

$$\begin{aligned}
\|A - B\| &= \lambda_{\max}(B - A) \\
&= \lambda_{\max}(A^{1/2}(A^{-1/2} B A^{-1/2} - I)A^{1/2}) \\
&\le \|A\|\lambda_{\max}(A^{-1/2} B A^{-1/2} - I) \\
&= \|A\|(\lambda_{\max}(A^{-1/2} B A^{-1/2}) - 1) \\
&= \|A\|(\exp(\delta_\infty(A, B)) - 1) \,.
\end{aligned}$$

This yields the first claim. The second follows from the crude bound that $e^x \le 1 + ex$ for $x \in (0, 1)$. $\qquad\square$

# E  Useful Perturbation Results

Here we collect various perturbation results which are used in Section B.2.

**Lemma E.1** (Lemma B.1, [35])**.** *Suppose that $K_0$ stabilizes $(A, B)$, and satisfies $\|(A + BK_0)^k\| \le \tau\rho^k$ for all $k$ with $\tau \ge 1$ and $\rho \in (0, 1)$. Suppose that $K$ is a feedback matrix that satisfies $\|K - K_0\| \le \frac{1-\rho}{2\tau\|B\|}$. Then we have that $K$ stabilizes $(A, B)$ and satisfies $\|(A + BK)^k\| \le \tau \mathsf{Avg}(\rho, 1)^k$.*

**Lemma E.2** (Lemma 1, [26])**.** *Let $f_1, f_2$ be two $\mu$-strongly convex twice differentiable functions. Let $x_1 = \arg\min_x f_1(x)$ and $x_2 = \arg\min_x f_2(x)$. Suppose $\|\nabla f_1(x_2)\| \le \varepsilon$, then $\|x_1 - x_2\| \le \frac{\varepsilon}{\mu}$.*

**Proposition E.3.** *Let $M \succeq \mu I$ and $N \succeq \mu I$ be a positive definite matrices partitioned as $M = \begin{bmatrix} M_{11} & M_{12} \\ M_{12}^\mathsf{T} & M_{22} \end{bmatrix}$ and similarly for $N$. Let $T(M) = -M_{22}^{-1} M_{12}^\mathsf{T}$. We have that:*

$$\|T(M) - T(N)\| \le \frac{(1 + \|T(N)\|)\|M - N\|}{\mu} \,.$$

*Proof.* Fix a unit norm $x$. Define $f(u) = (1/2)x^\mathsf{T} M_{11} x + (1/2)u^\mathsf{T} M_{22} u + x^\mathsf{T} M_{12} u$ and $g(u) = (1/2)x^\mathsf{T} N_{11} x + (1/2)u^\mathsf{T} N_{22} u + x^\mathsf{T} N_{12} u$. Let $u_\star = T(N)x$. We have that

$$\nabla f(u_\star) = \nabla f(u_\star) - \nabla g(u_\star) = (M_{22} - N_{22})u_\star + (M_{12} - N_{12})^\mathsf{T} x \,.$$

Hence, $\|\nabla f(u_\star)\| \le \|M_{12} - N_{12}\| + \|M_{22} - N_{22}\|\|u_\star\|$. We can bound $\|u_\star\| = \|T(N)x\| \le \|T(N)\|$. The claim now follows using Lemma E.2. $\qquad\square$

**Proposition E.4.** *Let $K, K_0$ be two stabilizing policies for $(A, B)$. Let $V, V_0$ denote their respective value functions and suppose that $V \preceq V_0$. We have that for all $k \ge 0$:*

$$\|(A + BK)^k\| \le \sqrt{\frac{\lambda_{\max}(V_0)}{\lambda_{\min}(S)}}(1 - \lambda_{\min}(V_0^{-1} S))^{k/2} \,.$$

*Proof.* This proof is inspired by the proof of Lemma 5.1 of Abbasi-Yadkori et al. [3]. Since $V$ is the value function for $K$, we have:

$$V = (A + BK)^\mathsf{T} V(A + BK) + S + K^\mathsf{T} RK$$
$$\succeq (A + BK)^\mathsf{T} V(A + BK) + S \,.$$

Conjugating both sides by $V^{-1/2}$ and defining $H := V^{1/2}(A + BK)V^{-1/2}$,

$$I \succeq V^{-1/2}(A + BK)^\mathsf{T} V(A + BK)V^{-1/2} + V^{-1/2}SV^{-1/2}$$
$$= H^\mathsf{T} H + V^{-1/2}SV^{-1/2} \,.$$

This implies that $\|H\|^2 = \|H^\mathsf{T} H\| \leq \|I - V^{-1/2}SV^{-1/2}\| = 1 - \lambda_{\min}(S^{1/2}V^{-1}S^{1/2}) \leq 1 - \lambda_{\min}(S^{1/2}V_0^{-1}S^{1/2})$. The last inequality holds since $V \preceq V_0$ iff $V^{-1} \succeq V_0^{-1}$, Now observe:

$$\|V^{1/2}(A + BK)^k V^{-1/2}\| = \|H^k\| \leq \|H\|^k \leq (1 - \lambda_{\min}(V_0^{-1}S))^{k/2}$$

Next, for $M$ positive definite and $N$ square, observe that:

$$\|MNM^{-1}\| = \sqrt{\lambda_{\max}(MNM^{-2}N^\mathsf{T} M)}$$
$$\geq \sqrt{\lambda_{\min}(M^{-2})\lambda_{\max}(MNN^\mathsf{T} M)}$$
$$= \sqrt{\lambda_{\min}(M^{-2})\lambda_{\max}(N^\mathsf{T} M^2 N)}$$
$$\geq \sqrt{\lambda_{\min}(M^{-2})\lambda_{\min}(M^2)\|N\|^2}$$
$$= \frac{\|N\|}{\kappa(M)} \,.$$

Therefore, we have shown that:

$$\|(A + BK)^k\| \leq \sqrt{\kappa(V)}(1 - \lambda_{\min}(V_0^{-1}S))^{k/2} \leq \sqrt{\frac{\lambda_{\max}(V_0)}{\lambda_{\min}(S)}}(1 - \lambda_{\min}(V_0^{-1}S))^{k/2} \,.$$

$\square$

**Proposition E.5.** *Let $A$ be a $(\tau, \rho)$ stable matrix, and let $\|\cdot\|$ be either the operator or Frobenius norm. We have that:*

$$\|\mathsf{dlyap}(A, M)\| \leq \frac{\tau^2}{1 - \rho^2}\|M\| \,. \tag{E.1}$$

*Proof.* It is a well known fact that we can write $P = \sum_{k=0}^\infty (A^k)^\mathsf{T} M(A^k)$. Therefore the bound follows from triangle inequality and the $(\tau, \rho)$ stability assumption. $\square$

**Proposition E.6.** *Suppose that $A_1, A_2$ are stable matrices. Suppose furthermore that $\|A_i^k\| \leq \tau\rho^k$ for some $\tau \geq 1$ and $\rho \in (0, 1)$. Let $Q_1, Q_2$ be PSD matrices. Put $P_i = \mathsf{dlyap}(A_i, Q_i)$. We have that:*

$$\|P_1 - P_2\| \leq \frac{\tau^2}{1 - \rho^2}\|Q_1 - Q_2\| + \frac{\tau^4}{(1 - \rho^2)^2}\|A_1 - A_2\|(\|A_1\| + \|A_2\|)\|Q_2\| \,.$$

*Proof.* Let the linear operators $F_1, F_2$ be such that $P_i = F_i^{-1}(Q_i)$, i.e. $F_i(X) = X - A_i^\mathsf{T} XA_i$. Then:

$$P_1 - P_2 = F_1^{-1}(Q_1) - F_2^{-1}(Q_2)$$
$$= F_1^{-1}(Q_1 - Q_2) + F_1^{-1}(Q_2) - F_2^{-1}(Q_2)$$
$$= F_1^{-1}(Q_1 - Q_2) + (F_1^{-1} - F_2^{-1})(Q_2) \,.$$

Hence $\|P_1 - P_2\| \leq \|F_1^{-1}\|\|Q_1 - Q_2\| + \|F_1^{-1} - F_2^{-1}\|\|Q_2\|$. Now for any $M$ satisfying $\|M\| \leq 1$

$$\|F_i^{-1}(M)\| = \left\|\sum_{k=0}^\infty (A_i^\mathsf{T})^k M A_i^k\right\| \leq \frac{\tau^2}{1 - \rho^2} \,.$$

Next, we have that:

$$\|F_1^{-1} - F_2^{-1}\| = \|F_1^{-1}(F_2 - F_1)F_2^{-1}\| \le \|F_1^{-1}\|\|F_2^{-1}\|\|F_1 - F_2\| \le \frac{\tau^4}{(1-\rho^2)^2}\|F_1 - F_2\|\,.$$

Now for any $M$ satisfying $\|M\| \le 1$,

$$\begin{aligned}
\|F_1(M) - F_2(M)\| &= \|A_2^\mathsf{T} M A_2 - A_1^\mathsf{T} M A_1\| \\
&= \|(A_2 - A_1)^\mathsf{T} M A_2 + A_1^\mathsf{T} M (A_2 - A_1)\| \\
&\le \|A_1 - A_2\|(\|A_1\| + \|A_2\|)\,.
\end{aligned}$$

The claim now follows. $\qquad\square$

## F   Useful Implicit Inversion Results

**Proposition F.1.** *Let $T \ge 2$ and suppose that $\alpha \ge 1$. Define $\varepsilon$ as:*

$$\varepsilon = \inf\left\{\varepsilon \in (0,1) : T \ge \frac{1}{\varepsilon^2}\log^\alpha(1/\varepsilon)\right\}\,,$$

*then we have*

$$\varepsilon \le \frac{\log^{(\alpha+1)/2}(T)}{\sqrt{T}}\,.$$

*As a corollary, if $T \ge 2C$ then if we define $\varepsilon$ as:*

$$\varepsilon = \inf\left\{\varepsilon \in (0,1) : T \ge \frac{C}{\varepsilon^2}\log^\alpha(1/\varepsilon)\right\}\,,$$

*then we have*

$$\varepsilon \le \sqrt{\frac{C}{T}}\log^{(\alpha+1)/2}(T/C)\,.$$

*Proof.* First, we know that such a $\varepsilon$ exists by continuity because $\lim_{\varepsilon \to 1^-} \frac{1}{\varepsilon^2}\log^\alpha(1/\varepsilon) = 0$.

Suppose towards a contradiction that $\varepsilon > \log^\beta(T)/\sqrt{T}$ where $2\beta = \alpha + 1$. Note that we must have $\log^\beta(T)/\sqrt{T} < 1$, since if we did not, we would have

$$1 \ge \varepsilon > \log^\beta(T)/\sqrt{T} \ge 1\,.$$

Therefore, by the definition of $\varepsilon$,

$$T < \frac{T}{\log^{2\beta}(T)}\log^\alpha(\sqrt{T}/\log^\beta(T)) \le \frac{T}{\log^{2\beta}(T)}\log^\alpha(\sqrt{T})\,.$$

This implies that:

$$\log^{2\beta}(T) \le \log^\alpha(\sqrt{T}) = \frac{1}{2^\alpha}\log^\alpha(T)\,.$$

Using the fact that $2\beta = \alpha + 1$, this implies:

$$\log(T) \le 1/2^\alpha \implies T \le \exp(1/2^\alpha) \le \exp(1/2)\,.$$

But this contradicts the assumption that $T \ge 2$.

The corollary follows from a change of variables $T \leftarrow T/C$. $\qquad\square$

**Proposition F.2.** *Let $C \ge 1$ and $\alpha \ge 1$. We have that:*

$$\sup_{i=0,1,2,\ldots} \frac{1}{(2^i)^{1/\alpha}}\operatorname{polylog}(C2^i) \le \operatorname{poly}(\alpha)\operatorname{polylog}(C)\,.$$

*Proof.* Let $\beta \geq 1$. We have that:

$$\frac{1}{(2^i)^{1/\alpha}}\log^\beta(C2^i) = \frac{1}{(2^i)^{1/\alpha}}(\log(C) + \log(2^i))^\beta$$

$$\leq \frac{2^{\beta-1}}{(2^i)^{1/\alpha}}(\log^\beta(C) + \log^\beta(2^i))$$

$$\leq 2^{\beta-1}\log^\beta(C) + 2^{\beta-1}\frac{i^\beta}{(2^i)^{1/\alpha}}\log^\beta(2) .$$

Next, we look at:

$$f(i) := \frac{i^\beta}{(2^i)^{1/\alpha}} .$$

We have that:

$$\frac{d}{di}\log_2 f(i) = \frac{\beta}{i \log 2} - \frac{1}{\alpha} .$$

Setting the derivative to zero we obtain that $i = \alpha\beta/\log 2$. Therefore:

$$\sup_{i=0,1,2,\ldots} f(i) \leq \beta\left(\frac{\alpha\beta}{\log 2}\right)^\beta .$$

The claim now follows. $\qquad\square$

**Proposition F.3.** *Let $C > 0$. Then for any $\varepsilon \in (0, \min\{1/e, C^2\})$, we have the following inequality holds:*

$$\varepsilon \log(1/\varepsilon) \leq C .$$

*As a corollary, let $M > 0$, then for $\varepsilon \in (0, \min\{M/e, C^2/M\})$ we have that:*

$$\varepsilon \log(M/\varepsilon) \leq C .$$

*Proof.* Let $f(\varepsilon) := \varepsilon \log(1/\varepsilon)$. We have that $\lim_{\varepsilon \to 0^+} f(\varepsilon) = 0$ and that $f'(\varepsilon) = \log(1/\varepsilon) - 1$. Hence $f$ is increasing on the interval $\varepsilon \in [0, 1/e]$, and $f(1/e) = 1/e$. Therefore, if $C \geq 1/e$ then $f(\varepsilon) \leq C$ for any $\varepsilon \in (0, 1/e)$.

Now suppose that $C < 1/e$. One can verify that the function $g(x) := 1/x + 2\log x$ satisfies $g(x) \geq 0$ for all $x > 0$. Therefore:

$$g(C) \geq 0 \Longleftrightarrow 1/C + 2\log C \geq 0$$

$$\Longleftrightarrow 1/C \geq \log(1/C^2)$$

$$\Longleftrightarrow C \geq C^2 \log(1/C^2)$$

$$\Longleftrightarrow f(C^2) \leq C .$$

Since $C < 1/e$ we have $C^2 \leq C$ and therefore $f(\varepsilon) \leq f(C^2) \leq C$ for all $\varepsilon \in (0, C^2)$. This proves the first part.

To see the second part, use the variable substitution $\varepsilon \leftarrow \varepsilon/M, C \leftarrow C/M$. $\qquad\square$

**Proposition F.4.** *Let $\beta \geq 1$ and $C \geq (e/\beta)^\beta$. Let $x$ denote the solution to:*

$$x = C\log^\beta(x) .$$

*We have that $x \leq e^{(\alpha-1)\beta}\beta^\beta \cdot C\log^\beta(\beta C^{1/\beta})$, where $\alpha = 2 - \log(e-1)$.*

*Proof.* Let $W(\cdot)$ denote the Lambert $W$ function. It is simple to check that $x = \exp(-\beta W(-\frac{1}{\beta C^{1/\beta}}))$ satisfies $x = C\log^\beta(x)$. From Theorem 3.2 of **?** ], we have that for any $t > 0$:

$$W(-e^{-t-1}) > -\log(t+1) - t - \alpha , \quad \alpha = 2 - \log(e-1) .$$

We now write:

$$
\begin{aligned}
W\left(-\frac{1}{\beta C^{1/\beta}}\right) &= W\left(-e^{\log\left(\frac{1}{\beta C^{1/\beta}}\right)}\right) \\
&= W\left(-e^{-\log(\beta C^{1/\beta})}\right) \\
&= W(-e^{-t-1}), \;\; t = \log(\beta C^{1/\beta}) - 1 \\
&> -\log(t+1) - t - \alpha .
\end{aligned}
$$

where the last inequality uses the result from Alzahrani and Salem and the assumption that $C \geq (e/\beta)^{\beta}$. We now upper bound $x$:

$$
\begin{aligned}
x &= \exp\left(-\beta W\left(-\frac{1}{\beta C^{1/\beta}}\right)\right) \\
&\leq \exp(\beta \log(t+1) + \beta t + \alpha\beta) \\
&= \exp(\beta \log\log(\beta C^{1/\beta})) \exp(\beta \log(\beta C^{1/\beta})) \exp((\alpha-1)\beta) \\
&= \exp((\alpha-1)\beta)\beta^{\beta} C \log^{\beta}(\beta C^{1/\beta}) .
\end{aligned}
$$

$\square$

## G  Experimental Evaluation Details

In this section we briefly describe the other algorithms we evaluate in Section 4, and also describe how we tune the parameters of these algorithms for the experiments.

Define the function $J(K; W)$ as:

$$
J(K; W) := \lim_{T \to \infty} \mathbb{E}\left[\frac{1}{T}\sum_{t=1}^{T} x_t^{\mathsf{T}} S x_t + u_t^{\mathsf{T}} R u_t\right] \tag{G.1a}
$$

$$
\text{s.t. } x_{t+1} = A x_t + B u_t + w_t, \;\; u_t = K x_t, \;\; w_t \sim \mathcal{N}(0, W) . \tag{G.1b}
$$

Certainty equivalence (nominal) control uses data to estimate a model $(\widehat{A}, \widehat{B}) \approx (A, B)$ and then solve for the optimal controller to (G.1) via the Riccati equations. On the other hand, both policy gradients and DFO are derivative-free random search algorithms on $J(K; W)$. For policy gradients, one uses action-space perturbations to obtain an unbiased estimate of the gradient of $J(K; \sigma_w^2 I + \sigma_\eta^2 B B^{\mathsf{T}})$. For DFO, random finite differences are used to obtain an unbiased estimate of the gradient of $J_{\sigma_\eta}(K) := \mathbb{E}_\xi[J(K + \sigma_\eta \xi; \sigma_w^2 I)]$, where each entry of $\xi$ is drawn i.i.d. from $\mathcal{N}(0, 1)$. Below, we describe each method in more detail.

**Certainty equivalence (nominal) control.**  The certainty equivalence (nominal) controller solves (G.1) by first constructing an estimate $(\widehat{A}, \widehat{B}) \approx (A, B)$ and then outputting the estimated controller $\widehat{K}$ via:

$$
\begin{aligned}
\widehat{K} &= -(\widehat{B}^{\mathsf{T}} \widehat{P} \widehat{B} + R)^{-1} \widehat{B}^{\mathsf{T}} \widehat{P} \widehat{A} , \\
\widehat{P} &= \mathsf{dare}(\widehat{A}, \widehat{B}, S, R) .
\end{aligned}
$$

The estimates $(\widehat{A}, \widehat{B})$ are constructed via least-squares. In particular, $N$ trajectories each of length $T$ are collected $\{x_t^{(i)}\}_{t=1, i=1}^{T, N}$ using the random input sequence $u_t^{(i)} \sim \mathcal{N}(0, \sigma_u^2 I)$, and $(\widehat{A}, \widehat{B})$ are formed as the solution to:

$$
(\widehat{A}, \widehat{B}) = \arg\min_{(A, B)} \frac{1}{2} \sum_{i=1}^{N} \sum_{t=1}^{T-1} \|x_{t+1}^{(i)} - A x_t^{(i)} - B u_t^{(i)}\|^2 .
$$

For our experiments, we set $\sigma_u = 1$.

**Policy gradients.** The gradient estimator works as follows. A large horizon length $T$ is fixed. A trajectory $\{x_t\}$ is rollout out for $T$ timesteps with the input sequence $u_t = Kx_t + \eta_t$, with $\eta_t \sim \mathcal{N}(0, \sigma_\eta^2 I)$. Let $\tau_{s:t} = (x_s, u_s, x_{s+1}, u_{s+1}, ..., x_t, u_t)$ denote a sub-trajectory, and let $c(\tau_{s:t})$ denote the LQR cost over this sub-trajectory, i.e. $c(\tau_{s:t}) = \sum_{k=s}^{t} x_k^\mathsf{T} S x_k + u_k^\mathsf{T} R u_k$. The policy gradient estimate is:

$$\widehat{g} = \frac{1}{T} \sum_{t=1}^{T} \frac{c(\tau_{t:T})}{\sigma_\eta^2} \eta_t x_t^\mathsf{T} \ .$$

Of course, one can use a baseline function $b(\tau_{1:t-1}, x_t)$ for variance reduction as follows:

$$\widehat{g} = \frac{1}{T} \sum_{t=1}^{T} \frac{c(\tau_{t:T}) - b(\tau_{1:t-1}, x_t)}{\sigma_\eta^2} \eta_t x_t^\mathsf{T} \ .$$

**DFO.** We use the two point estimator. As in policy gradients, we fix a horizon length $T$. We first draw a random perturbation $\xi$. Then, we rollout one trajectory $\{x_t\}_{t=1}^{T}$ with $u_t = (K + \sigma_\eta \xi)x_t$, and we rollout another trajectory $\{x_t'\}_{t=1}^{T}$ with $u_t' = (K - \sigma_\eta \xi)x_t'$. We then use the gradient estimator:

$$\widehat{g} = \frac{\frac{1}{T}\sum_{t=1}^{T} c_t - \frac{1}{T}\sum_{t=1}^{T} c_t'}{2\sigma_\eta} \xi \ , \quad c_t = x_t^\mathsf{T} S x_t + u_t^\mathsf{T} R u_t \ , \quad c_t' = (x_t')^\mathsf{T} S x_t' + (u_t')^\mathsf{T} R u_t' \ .$$

**MFLQ.** We update the policy every 100 iterations and do not execute a random exploratory action since we found that it negatively affected the performance of the algorithm in practice. In terms of the parameters described in Algorithm 1 of Abbasi-Yadkori et al. [3] we execute v2 of the algorithm and set $T_s = \infty$ and $T_v = 100$. We also chose to use all data collected throughout an experiment when updating the policy.

**Optimal.** The optimal controller simply solves for the optimal controller to G.1 given the true matrices $A$ and $B$. That is, it uses the controller

$$K = -(B^\mathsf{T} P B + R)^{-1} B^\mathsf{T} P A \ ,$$
$$P = \mathsf{dare}(A, B, S, R) \ .$$

**Offline setup details.** Recall that we use stochastic gradient descent with a constant step size $\mu$ as the optimizer for both policy gradients and DFO. After every iteration, we project the iterate $K_t$ onto the set $\{K : \|K\|_F \leq 5\|K_\star\|_F\}$, where $K_\star$ is the optimal LQR controller (we assume the value $\|K_\star\|_F$ is known for simplicity). We tune the parameters of each algorithm as follows. We consider a grid of step sizes $\mu$ given by $[10^{-3}, 10^{-4}, 10^{-5}, 10^{-6}]$ and a grid of $\sigma_\eta$'s given by $[1, 10^{-1}, 10^{-2}, 10^{-3}]$. We fix the rollout horizon length $T = 100$ and choose the pair of $(\sigma_\eta, \mu)$ in the grid which yields the lowest cost after $10^6$ timesteps. This resulted in the pair $(\sigma_\eta, \mu) = (1, 10^{-5})$ for policy gradients and $(\sigma_\eta, \mu) = (10^{-3}, 10^{-4})$ for DFO. As mentioned above, we use the two point evaluation for derivative-free optimization, so each iteration requires $2T$ timesteps. For policy gradient, we evaluate two different baselines $b_t$. One baseline, which we call the *simple* baseline, uses the empirical average cost $b = \frac{1}{T}\sum_{t=1}^{T} c_t$ from the previous iteration as a constant baseline. The second baseline, which we call the *value function* baseline, uses $b(x) = x^\mathsf{T} V(K)x$ with $V(K) = \mathsf{dlyap}(A + BK, S + K^\mathsf{T} RK)$ as the baseline. We note that using this baseline requires exact knowledge of the dynamics $(A, B)$; it can however be estimated from data at the expense of additional sample complexity (c.f. Section 2.1). For the purposes of this experiment, we simply assume the baseline is available to us.

**Online setup details.** In the online setting we warm-start every algorithm by first collecting 2000 datapoints collected by feeding the input $Kx_t + \eta_t$ to the system where $K$ is a stabilizing controller and $\eta_t$ is Gaussian distributed aditive noise with standard deviation 1. We then run each algorithm for $10,000$ iterations. In the case of LSPI we set the initial number of policy iterations $N$ to be 3 and subsequently increase it to 4 at 2000 iterations, 5 at 4000 iterations, and 6 at 6000 iterations. We also follow the experimental methodology of Dean et al. [13] and set $T_i = 10(i + 1)$ and set $\sigma_{\eta,i}^2 = 0.01\left(\frac{1}{i+1}\right)^{2/3}$ where $i$ is the epoch number. Finally we repeat each experiment for 100 trials.

## G.1 Additional Experiments

In this section we include the results of additional experiments we ran in the online setting. We ran the nominal, LSPI, and MFLQ algorithms in the same online setting described previously, keeping track of relative error $(J(\widehat{K}) - J_\star)/J_\star$ between the current controller and the optimal controller.

**Figure 2:** The cost suboptimality of MFLQ, LSPI, and the nominal controller when compared with the baseline of the optimal controller in the adaptive setting. The shaded regions represent the median to upper 90th percentile over 100 trials. Here, LSPI is Algorithm 3 using LSPIv1, MFLQ is from Abbasi-Yadkori et al. [3], nominal is the $\varepsilon$-greedy adaptive certainty equivalent controller (c.f. [13]), and optimal has access to the true dynamics.

As Figure 2 shows, both LSPI and MFLQ perform similarly with LSPI slightly outperforming MFLQ towards the end of the experiment. Nominal significantly outperforms both model-free algorithms.