[Reviews · NeurIPS 2019]

Reviewer 1



*** Thanks. I've read the responses, which answered my questions. *** This paper is clearly written and the results are insightful. The theoretical results rely on non-trivial combinations of several recent techniques. I enjoy reading the paper and appreciate the contribution. However, I must admit that I am not particularly familiar with the field on RL+LQR, so I cannot comment on the details of the derivations. Below are some questions I had after reading the paper. 1. Projection step Algorithm 1 and 2 both use a Proj_\mu to project the Q matrix onto pd matrices. I am wondering if this is truly necessary or could potentially hurt performance. It seems from (2.10), we do not need Q to be pd. Though the optimal Q is pd, how do we estimate \mu in practice? 2. Adaptive algorithm First, LSPlv2 is not clearly defined in the main text, though it can be understood from the appendix. My concern is that, it is unclear whether K^{i+1} will remain to be stable in this online setup. If not, wouldn't it violate the stable behavior policy assumption? The proof in appendix appear to rely on Assumption 1, which seems pretty strong, because wouldn't that imply the optimal policy is almost solved in one online iteration? 3. Practicality Though this paper is theoretically focused, but from a practical perspective, I am wondering if the model-free setting is a reasonable approach to LQR. This is not a criticism, but as the paper states, it has been shown model-based is superb both theoretically and empirically. I am guessing, when would model-free approaches start to shine. For example, would model-free approaches start to make more sense, when the state is high dimensional and without low-rank structure (or even infinite dimensional)? while the action space is low-dimensional. The experiments considered are fairly simple. I think having some motivating problems would help readers to better appreciate technical contributions here. Minor: In line 73, K_eval is not defined, though it can be understood as the feedback gain from the later context.

Reviewer 2



1. Originality: while the algorithm studied in the paper is standard (i.e., approximate policy iteration and LSPI), I think the sample complexity results are new for LQR setting. Previous work on approximate PI on LQR is for deterministic setting and only studied asymptotic convergence. 2. Quality & Clarity: I think the paper is well written and organized. The survey on related work is thorough as well. 3. While I enjoyed reading the paper and analysis, I'm not quite certainty about the significance of the results presented in the paper. The proposed analysis seems not that straightforward to be extended and be leveraged to analyze more general settings (in general, achieving a uniformly accurate value function is also hard, which could make naive policy improvement unstable). Regarding the practical performance, the model-based (certainty equivalence method) clearly dominate the performance. The certainty equivalence method itself is straightforward and is recently proved to be efficient as well. It seems to me that when I'm trying to solve LQR learning problem, in default I would use the nominal control approach as planning in LQR is straightforward, which leaves the question when/under what circumvents I would use the proposed LSPI algorithm? Some discussion on the significance of the results will be useful.

Reviewer 3



SUMMARY: The paper considers the Linear Quadratic Regulator (LQR) design problem when the model is not known, and we only have access to data. It is a simple form of an RL problem with a particular dynamics and reward (or cost) function. The paper considers the sample complexity of using a Least-Squares Policy Improvement (LSPI) algorithm to solve the LQR problem. It provides a finite-sample guarantee on the quality of the learned value function for the policy evaluation part of LSPI (i.e., LSTD-Q) in Theorem 2.1. It then considers the LSPI procedure and provides a guarantee on the quality of the learned policy (which is the gain of a linear controller in this case) in Theorem 2.2. These results are for the case when the data is generated by a fixed policy (specified by K_play). The paper considers the adaptive version of LQR and proves a regret bound (Theorem 2.3). Finally, some empirical results are reported in which a model-based approach, which estimates the model and uses the learned model to design the controller, generally outperform the model-free methods, including the LSPI approach. EVALUATION: [BRIEF] Originality: The algorithm and type of analysis are not particularly novel, but the theoretical result is. Quality: The claims seems to be rigorously proven. I have some questions, including whether Theorem 2.2 upper bounds the right quantity ( || K_N - K_* || instead of the error in the value functions). Clarity: The paper is written clearly. Significance: The theoretical result is important for better understanding a class of RL methods for solving a particular class of RL algorithms. It might provide insights for more general class of RL problems too. This is a well-written paper that closely studies LSPI in the context of LQR problem. The LQR problem is interesting on its own, and hopefully the results from analyzing this simpler problem can be insightful for more general RL problems. I skimmed through the proofs, but I did not verify them closely. As far as I can say, especially from sampling a few points of the argument here and there, the authors seem to be meticulous in proving their results. I have some questions and comments. I use [*], [**], etc. in order to show the importance of my comments, in case the authors have to prioritize their answers. - [**] Is it correct that the control signal u_t is noisy for both K_play and K_eval? If so, does it mean that q in Theorem 2.1 (specifically in Eq. 2.9) reflects the value of the stochastic policy and not the deterministic policy that is defined by K_eval? If it is the case, why do we need noise in K_eval? I understand that K_play may need noise for exploration (even though it is not obvious that it is indeed required, as the dynamics is already noisy and might provide the necessary persistent excitation). It is not clear, however, why it is needed for K_eval? My guess is that the value of the stochastic policy is in fact worse than the value of the deterministic policy. - [**] The vector q and \hat{q} in Theorem 2.1 describe the parameters of the linear function approximator with the feature map phi. For the LQR problem, the value function is quadratic in state and actions (as specified at L79). My question is whether phi is chosen to be of quadratic form for the result of Theorem 2.1 to hold? Or is it true for any phi? A related question: If phi is not in quadratic form (so Q(x,u) = phi(x,u) q may not be able to represent the true value function), what does q in Theorem 2.1 refer to? Is it the project of the true value function onto the span of phi? - [**] If we let sigma_w go to zero, the upper of Eq. 2.9 approaches infinity. Is this what we expect? Does this mean that learning to control a (close to) deterministic system is very difficult? - [***] The result of Theorem 2.2 is about the closeness of K_N to K_* (with K being the gain of the controller). How does this translate to the difference between the value functions? (I understand that the value function is a quadratic function of state, so it is unbounded, but we can still talk about the difference in the value functions, for example, within a unit ball; or upper bound the error in P, as is done in Theorem 4.3 of [36] Tu and Recht, 2018). - [*] A related issue is that comparing the result of Theorem 2.2 with a result such as the upper bound of L160 is not completely fair, not only because one is about discounted setting and the other is for undiscounted one (as correctly claimed), but also because one provides a guarantee on the value function and the other (this paper) only on the policy. - [**] Another point of difference is the aforementioned result is an L_inft bound. There are results in the L_p too. Some examples: - Munos, “Error Bounds for Approximate Policy Iteration,” ICML, 2003. - Farahmand, Munos, Szepesvari, “Error Propagation for Approximate Policy and Value Iteration,” NeurIPS, 2010. - Scherrer, Ghavamzadeh, Gabillon, Geist, “Approximate Modified Policy Iteration,” ICML, 2012. As well as the followings cited ones: [6] Antos et. al. [17] Farahmand et al. [23] Lazaric et. al - [*] As a side note, the theoretical analysis of Farahmand et al. [17] is not specialized to RKHS; it is for general nonparametric function spaces. The closed-form formulae, however, are. - [*] What is \tilde{x} in L377? Is it the next state? - [*] L415: Here we have E[||phi_t||^2] = E[||z_t||^4]. Is this because the choice of feature map is a quadratic function? - [*] L762: Where is Avg(rho,1) defined? Is it simply the average of 1 and rho? - [*] The term “state-value”, instead of “action-value”, is used to refer to Q at several places, including on L.532, 544, 545. - [**] By now we have several papers studying the finite-sample properties of solving an LQR, either in the model-based or model-free setting. What are the insights that we have gained from solving this simpler problem that might be useful for more general RL problems? A discussion on this would be very helpful for the paper. - [**] (This might be a bit subjective) It is claimed in LL25-28 that “On the other hand [comparison is in contrast with the work on LQR], for classic model-free RL algorithms such as Q-learning, SARSA, approximate policy iteration (PI), our understanding is much less complete despite the fact that these algorithms are well understood in the tabular setting”. Even though there is much remained to be understood about these algorithms, especially for the large state and action spaces, I do not believe that they are significantly less understood compared to the LQR setting (especially in terms of the finite-sample type of analysis, and not how to solve them as solving LQR is relatively easy). This has paper already cited a few related work such as [6] Antos et. al., [17] Farahmand et al., [23] Lazaric et. al., and there are several others too. It is true that those results do not generally provide sharp constants. But then, this work does not either. For example, Theorem 2.2 of this work has a problem-dependent constant that is raised to the power of 42! So the bound is technically very loose. I understand that large constants might appear in the bounds and if the subject of study is the behaviour of other terms (say, the dependence on eps or dimensions), it is often acceptable to have loose dependence on other terms.

[Author Response · NeurIPS 2019]

We thank all the reviewers for their helpful comments and suggestions which will substantially improve our manuscript.
Before we respond to specific comments, we want to address a high level concern which was brought up by several
reviewers regarding what we learn about RL from studying finite sample bounds for LQR. We note that most of the prior
work in LQR has been focused on model-based methods. Motivated by the popularity of model-free methods in practice,
our goal was to compare classic model-free algorithms and see how they measure up against the model-based methods
on LQR. The results in this paper suggest that there is a substantial sample complexity hit when using model-free
methods. We believe there are two high level takeaways: (a) we give a theoretical understanding of the trade-offs
between policy evaluation and policy improvement for LQR, which is a core theoretical question in RL, and (b) we
believe that there are small deltas to the problem setup studied in this paper which might result in model-free algorithms
being more competitive with model-based algorithms, such as partial observability or introducing simple non-linearities;
it is conceivable that analyzing model-free methods with these small deltas could heavily build on our tools and analysis.
Finally, we hope that this line of research motivates further study into when to use model-based methods vs. model-free
methods for more general RL.

**Reviewer #2.**   Regarding the projection step, in practice we find that the projection step improves performance by
helping to stabilize the algorithm in the beginning iterations. While the update defined by (2.10) would technically
work as long as $Q_{22}$ is invertible, a greedy policy improvement step with respect to a quadratic value function that is
not positive definite is not well-defined.

Regarding the quantity $\mu$, we note that $\mu$ does not actually need to be estimated since we simply set it to the minimum
of $\lambda_{\min}(S)$ and $\lambda_{\min}(R)$, and the cost matrices are assumed to be known.

Next, we remark that LSPIv2 is defined in the main text in Algorithm 2 that starts on Line 122.

Finally, regarding whether $K^{(i+1)}$ is stable, we note that a non-trivial part of the regret proof is dedicated to ensuring
that $K^{(i+1)}$ is stable. Assumption 1 is the main abstraction we use that allows us to analyze the meta-algorithm in
Algorithm 5. It says that the batch learning algorithm EstimateK we use has a $O(\varepsilon)$ guarantee on sub-optimality using
$O(1/\varepsilon^2)$ samples. The point of the online algorithm is to allow us to use any batch learning method which satisfies
this guarantee and send $\varepsilon$ to zero in a way that incurs sub-linear regret. Using Assumption 1, we carefully work
through the perturbation analysis (c.f. perturbation related results in Section E) to ensure that enough data is collected
to maintain stability. Finally, we show in Lines 718-724 that LSPIv2 satisfies Assumption 1 (by Theorem 2.2) and make
the constants $C_{\mathrm{req}}, C_{\mathrm{err}}$ explicit.

**Reviewer #4.**   We hope that our comments in the first paragraph address the main concerns with our work.

**Reviewer #5.**   Regarding if the control signal $u_t$ is noisy for both $K_{\mathrm{play}}$ and $K_{\mathrm{eval}}$, the $q$ parameter in Theorem 2.1
reflects the parameter of the Q-function in (2.3) where the policy $\pi(x) = Kx$. It is correct that $K_{\mathrm{play}}$ uses noise for
exploration, but $K_{\mathrm{eval}}$ is not considered a stochastic policy (c.f. the definition of $\psi_t$ in Line 86). We will make this more
clear in the writing. As for exploration noise in $K_{\mathrm{play}}$, if $K_{\mathrm{play}}$ had no exploration noise then the associated covariance
matrices for LSTD-Q would be degenerate.

Regarding the generality of Theorem 2.1, Theorem 2.1 only applies to the situation where the data is actually coming
from an LQR system and hence the solution to the Bellman equation actually has a quadratic function solution. It is not
a general result for LSTD-Q, but specific to the estimator (2.5).

Regarding letting $\sigma_w$ to zero, if we let $\sigma_w$ go to zero, then the RHS of (2.9) goes to zero, meaning there will be perfect
estimation of $q$. This is to be expected, since in a deterministic system we should be able to recover exactly the $q$
parameter after $O((n+d)^2)$ samples (as long as the covariates are non-singular). Perhaps the reviewer meant to refer
to sending $\sigma_\eta$, the exploration parameter, to zero? In which case yes, if we do not properly excite the system, then we
should not expect to be able to recover the $q$ parameter (the associated covariance matrices will be denegerate).

Regarding closeness of $K_N$ to $K_\star$, please refer to Lemma C.3 (which is Lemma 12 in Fazel et al.), which relates the
error in the controller to the error in the value functions. It says that an $O(\varepsilon)$ error in $\|K - K_\star\|$ translates to an $O(\varepsilon^2)$
error in the value functions. In this paper, we focus on the sub-optimality gap $J(K) - J_\star$, which is the main quantity of
interest in prior work for model-based LQR (e.g. Dean et al). The reviewer is correct to point out that this is not quite
the same as focusing on $\|P - P_\star\|$ as in Tu et al.

Regarding $L_\infty$ bounds on value functions, thanks for bringing the papers to our attention. We will remark about more
refined $L_p$ bounds on value functions and include the relevant citations in our revision.

Regarding the comment about Lines 25-28, I think our wording here is unfortunately a bit unclear. What we meant to
say was that the behavior of model-free methods on LQR is much less well understood compared to the behavior of
model-based methods on LQR, and this paper is an attempt to address this. We will clean up this phrasing.

[Meta-Review · NeurIPS 2019]

The paper studies approximate policy iteration methods in LQR models, contributing to the theoretical results for model-free methods in this setting. The finite-sample results are new and interesting. One issues raised by reviewers is whether this analysis would be generalizable to other more complex RL settings